# Biomarkers of professional cybersportsmen: Event related potentials and cognitive tests study

**Sergei Gostilovich**[ID]<sup></sup>*, **Airat Kotliar Shapirov**<sup></sup>, **Andrei Znobishchev**<sup></sup>, **Anh-Huy Phan**, **Andrzej Cichocki**

Center for Artificial Intelligent Technology, Skolkovo Institute of Science and Technology (Skoltech), Moscow, Russia

☯ These authors contributed equally to this work.

* sergei.gostilovich@gmail.com

**Data Availability Statement:** ***PA @ ACCEPT: please ask for additional data at accept*** Partial data will be available online at the following link: https://disk.yandex.ru/d/jgW45J3tJKUAWw.

## Abstract

"Faster, higher, stronger" is the motto of any professional athlete. Does that apply to brain dynamics as well? In our paper, we performed a series of EEG experiments on Visually Evoked Potentials and a series of cognitive tests—reaction time and visual search, with professional eSport players in Counter-Strike: Global Offensive (CS:GO) and novices (control group) in order to find important differences between them. EEG data were studied in a temporal domain by Event-Related Potentials (ERPs) and in a frequency domain by Variational Mode Decomposition. The EEG analysis showed that the brain reaction of eSport players is faster (P300 latency is earlier on average by 20-70 ms, $p < 0.005$) and stronger (P300 peak amplitude is higher on average by 7-9 mkV, $p < 0.01$). Professional eSport players also exhibit stronger stimulus-locked alpha-band power. Besides, the Spearman correlation analysis showed a significant correlation between hours spend in CS:GO and mean amplitude of P200 and N200 for the professional players. The comparison of cognitive test results showed the superiority of the professional players to the novices in reaction time (faster) and choice reaction time—faster reaction, but similar correctness, while a significant difference in visual search skills was not detected. Thus, significant differences in EEG signals (in spectrograms and ERPs) and cognitive test results (reaction time) were detected between the professional players and the control group. Cognitive tests could be used to separate skilled players from novices, while EEG testing can help to understand the skilled player's level. The results can contribute to understanding the impact of eSport on a player's cognitive state and associating eSport with a real sport. Moreover, the presented results can be useful for evaluating eSport team members and making training plans.

## Introduction

The digital era truly altered the world around us: robots automate many routine processes, the internet connects people all around the globe, machine learning models already solve some problems better than humans, and the rate of these changes is becoming only more rapid.

Access to the full data can be requested via the e-mails of the authors. More information is provided in the ReadMe file available at the specified link.

**Funding:** The authors received no specific funding for this work.

**Competing interests:** The authors have declared that no competing interests exist.

Besides, there are novel areas of activity, which are fully based on digital technologies. Such sphere is the focus of this work, i.e., cybersports or eSports: millions of people from all around the world compete with each other online in video games for entertainment, glory, and high prizes.

Video games can be of great interest to science since the game industry produces large amounts of diverse data that can be used for thorough analysis. One of the most popular and engaging game genres is "First-person shooter" (FPS), where a player controls the movement of a game character and sees the virtual world through his eyes. Moreover, his character's weaponry is used to combat other players. Other rules are game-specific. FPS genre games require players to have various skills and abilities in order to achieve high results. Scientific research in this new area is still in the early steps and many questions remain unanswered.

The aim of the current study is to identify the unique characteristics of professional cyber-athletes. Specifically, the abilities of players of the video game called Counter-Strike: Global Offensive (CS:GO) were studied. Counter-Strike is one of the most popular FPS games, which was developed by Valve Corporation in 1999. The most recent version is called Counter-Strike: Global Offensive and, according to Steam statistics, is being played by hundreds of thousands of people daily. Main rules of CS:GO are the following: Terrorists and Counter Terrorists teams compete with each other in order to achieve their opposing goals. Terrorists aim at planting a bomb, while Counter Terrorists try to defend a location, defuse the bomb, and rescue hostages. After each round, players get some amount of virtual money based on their performance, which they could spend on weapons for the next rounds. More money is given to a team that wins a round. Furthermore, additional bonuses are given to players who killed opponents' characters. Killing teammates is penalized. The team that wins some pre-defined number of rounds wins the game.

## Neural and cognitive correlates in sports

Despite the growing interest in video games, only a few studies exist that cover the topic of video game players' expertise and skills. Since many of them borrow ideas from studies on professional athletes, it is worth reviewing this area first. A plethora of papers is devoted to finding a set of abilities of professional athletes of a specific kind of sport. For example, the study [1] examined 50 arm sports athletes, dividing them into two groups by their experience, and observed a correlation between their proficiency and the level of carpal dynamometry. Sensory data is typically used to find unique patterns in the neuromuscular or brain activity of professional sportsmen. In [2], the authors investigated electromyography (EMG), the electrical activity of muscle tissue, of lower and upper limbs in soccer, handball, basketball, and volleyball players. The study concluded that EMG amplitude of the gastrocnemius muscle and several other features corresponding to lower limbs increase in handball, basketball, and volleyball athletes. Besides, other studies assess neural correlates of professional sportsmen. Another research [3] obtained psycho-physiological and behavioral indicators to assess the degree of anticipation of wrestlers using simultaneous recording of oculomotor activity and registration of multichannel electroencephalogram (EEG). EEG signals are widely used for identifying EEG biomarkers that can be of relevance to sport, as was mentioned in [4]. According to this review paper, both temporal and frequency domains were explored in various studies.

One of the widely used methods for studying EEG biomarkers in a temporal domain is to record the event-related potentials (ERPs). This approach allows to study reaction and decision-making skills more thoroughly, as was shown in [5]. The authors suggested that early ERP components (P100, N100, P200, N200) provide information about unconscious reactions,

while later components give insights into subjective awareness. ERP approach was implemented in many studies according to this review [6]. As an example, the authors describe a study where the combination of classical factors (3D movements and EMG) and the ERPs recording was applied for human performance evaluation. It was found [7] that subjects who are more engaged in physical exercises have larger P300 amplitude. In [8, 9], the authors recorded ERPs to study brain responses to auditory stimuli and discovered significant differences in ERP peak amplitudes for different groups. Furthermore, ERPs were measured to study motor readiness, visual attention, and reaction time of 15 elite table tennis players and 15 control subjects [10]. They found that table tennis players had superior reactivity to stimuli occurring in uncertain positions. A similar study was performed for skilled and less-skilled cricket batsmen. It was observed [11] that P300 latency and amplitude were significantly different for the two groups, which suggested the superior perceptual decision-making ability of skilled sportsmen. Studying ERPs is an extensively used technique for identifying neural correlates. It was found [12] that early ERP components (P100 and P200) showed amplitudes differences between target and non-target stimuli for professional martial arts athletes. Another research [13] aimed to find neural correlates to badminton players' anticipation abilities. Predicting the trajectory of a ball is a crucial skill in the game. Thus, the EEG signal recorded while professional players and non-players were watching videos of badminton games was analyzed. The authors observed differences in various ERP components (P300 and P200), indicating superior action anticipation intrinsic to professional badminton players. ERPs allowed to conclude that elite karate athletes have improved sustained visuo-spatial attention and self-control [14].

Several studies dedicated to cognitive function analysis, in turn, took advantage of transforming neural data into the frequency domain. The study [15] describes how alpha (8–12 Hz) and theta (4–8 Hz) oscillations in EEG signals can be associated with cognitive and memory performance. Theta power was found to reflect the encoding of new information, while alpha frequency was connected to search and retrieval processes in long-term memory. Similar findings published in [16] showed that subjects with higher IQ had larger alpha power during information processing. According to [17], alpha power decreased in the frontal and occipital regions of the brain when subjects were engaged in attention tasks. In [18, 19], the authors suggested that power in theta and alpha bands is a reliable biomarker of relaxation of the central nervous system. The work [20] revealed dissimilarities in power spectra between golfers of different expertise levels. Novice golfers were observed to have lower midline theta power, which is correlated with the ability to focus on a specific task. Besides, the research [21] studied the EEG alpha power of professional shooters and discovered that the alpha power increased before the best shots and decreased before the worst shots.

Another direction is studying various mental or cognitive attributes, which can serve as indicators of the expertise level of athletes. In [22], the authors assessed function state of martial arts athletes and ranked kick-boxers, karate, taekwondo, and wrestling sportsmen based on motor skills, choice reaction time, and a line's speed. The level of stability to deceiving signals in the conducted experiments did not identify statistically significant differences among them. Professional footballers demonstrated the beyond mid-high level of problem-solving skills, according to [23]. The same skill was found to be important for professional orienteering athletes [24]. The research [25] assessed elite and sub-elite tennis players on working memory, inhibitory control, and cognitive flexibility tests and detected their performance to be above the norm. Moreover, some of the results indicate differences between elite and sub-elite sportsmen (elite players had a lower rate of mistakes on tests for inhibitory control).

## Cognitive and neural correlates in video games

Moving on to the field of video games, a systematic review paper should be noted in particular [26]. The authors used standardized search operations in order to find and categorize articles related to video games, players' cognitive factors, and neuro-imaging techniques. One hundred studies provided functional data, and 22 measured structural brain changes. However, not all of them focused on the peculiar characteristics of elite players. For example, one-third of them covered video game addiction, and 14% focused on video game-related violence. A wide range of articles devoted to the unique attributes of video game players is dedicated to their cognitive abilities. [27] serves as an example of such articles. In a similar manner as karate athletes were compared with kick-boxers, researchers found differences between action video game players (AVGPs) and non-action video game players (NAVGPs). AVGPs were found to have enhanced perceptual and attentional control functions. Besides [28], studied the differences in reaction time of professional and non-professional eSports players from different genres and non-professional traditional athletes. The comparison of the groups itself does not detect significant differences. However, the study showed that payers in FPS had faster reaction time than players in Multiplayer Online Battle Arena (MOBA) games. In other words, the influence of game genre on players' cognitive skills was discovered. In [29], the role of a video game genre also was detected. During this study, the two groups of video game players (VGPs) of specific genres (first-person shooter and real-time strategy (RTS)) and non-video game players (NVGPs) were compared by task switching and multiple-object tracking tests. Both groups of VGPs showed better task switching performance than NVGPs. However, the RTS players were superior to both other considered groups in multi-object tracking. Some other studies were focused on the game genre itself. For instance, in [30], the researchers have studied how FPS games affect cognitive flexibility. The authors compared video-game players and subjects with no experience in video games (or NVGPs) and concluded that VGPs have smaller switching costs, thus, demonstrating better cognitive flexibility. Moreover, in [31], the comparison between VGPs and NVGPs showed that video-game players are faster in reacting to go signals but have comparable stopping performance. In addition, the study also claimed that VGPs also had superior working memory. Besides, in [32], it was suggested that VGPs have better visual attention and executive functions.

Other studies focused on video game players and the characteristics related to EEG. In [33], for example, the authors suggested a method to determine players' immersion in a computer game. The degree of engagement is proposed to be measured by their reaction to irrelevant stimuli presented during the game. Thus, 20 participants played a video game and were presented with auditory stimuli unrelated to the game. Their ERPs were measured along with obtaining The Immersive Experience Questionnaire, which was used to get subjective indicators of players' immersion. Results show that ERPs related to irrelevant stimuli can serve as a reliable marker of subjects' video game immersion. However, further studies should be carried out in order to investigate the hypothesis that professional players are more or less immersed in the game than non-players. The study [34], in turn, demonstrated that cognitive workload during playing video games can be reliably assessed through behavioral and neuroelectric means. The authors showed that single sound signals during playing the FPS game evoked less amplitude of P200, N200, and P300 ERP components as the gaming workload increased. In addition, there are several studies dedicated to the comparison between video game players and non-players based on EEG signals. The work [35] considered temporal domain and found the higher amplitude of P100 and P200 ERP components for video game players than non-players. The authors of [36] revealed the superior coupling between the prefrontal and parietal cortices of video game players based on increased right intrahemispheric prefrontal-parietal

correlation in the gamma band. Finally, the article [37] was dedicated to the League of Legends game (MOBA game) specifically. The authors used behavioral tests and EEG measurements to identify indices of multiple-object tracking capability, mental concentration, and visuospatial attention, which were different between professional, semi-professional, and novice players. The authors also reported that, for the behavioral tests, the statistically significant differences between the considered groups were detected only for multi-object tracking but not for reaction time and visual search skills.

We would like to mention the possible downside of video games: being developed as leisure activities they still pose a risk to turn into addictive behavior, which has been coined as Gaming Disorder (GD). The authors of [38] report that GD shares neurobiological alterations that are typical for other addictions such as "(i) activation in brain regions associated with reward, as evident from cue exposure and craving studies and neurotransmitter systems studies that indicate an involvement of dopamine-mediated reward mechanisms; (ii) reduced activity in impulse control areas and impaired decision-making; and (iii) reduced functional connectivity in brain networks that are involved in cognitive control, executive function, motivation, and reward". GD is also being reported to alter EEG oscillations [39] results suggest individuals with GD have raised delta and theta activity and reduced beta activity, with coherence analysis suggesting altered brain activity in the mid-to-high frequency range. Moreover, addictive behavior aside, video game play induces strong theta activity, and the frontal midline theta-wave activity increases over time relative to the eye open resting condition. These results suggest that long, continuous periods of video game play increase the mental load on the players, which may have mental health implications [40]. One could argue that it simply indicates engagement in the process just as in regular sports [14]. However, since the time spent in the virtual game is less bound by muscle stamina, it poses a bigger risk of mental overload.

## Motivation

Counter-Strike is a fairly complicated game typically requiring players thousands of hours of training to become professional. It is of great interest to investigate what kind of traits are specific to elite players of CS:GO. The main focus of the present research is finding dissimilarities in the cognitive abilities of players belonging to different skill groups. While some studies suggest continuous recording of EEG signals when subjects are engaged in a game, we propose to extract only key events from the game and record EEG while subjects are presented with visual stimuli related to the identified key events. More concretely, the presence of an enemy game character was chosen as the key event, and the ERPs provoked by these events were studied in the current research. There are two main reasons why the described approach seems to be more reasonable. Firstly, the CS:GO game environment continuously presents a subject with a variety of auditory and visual stimuli. The reaction of the brain to such stimuli is extremely complex and intricate. Besides, the lack of game data annotation complicates the analysis of the obtained signal. Secondly, ERP is a well-established approach to finding unique expertise features, as described above. Thus, the current study is based on a solid foundation of research conducted in a sports domain and aims to expand it to a novel area of eSports.

In addition to studying the participants' brain signals, it is also interesting to explore the cognitive functions which have links to the conducted EEG experiments. In our case, the participants need to see the pictures (reaction time), find an enemy game character (visual search), and make a choice if the picture is a target picture (decision-making). Therefore, reaction time, visual search, and decision-making were tested. We considered decision-making

together with reaction time as a choice reaction time. In total, two tests on reaction time, one test on choice reaction time, and one on visual search were conducted during the current research.

Speaking more about the mentioned cognitive skills, it should be noted that Counter-Strike is related to fast-changing and dynamic actions, which require fast and correct responses to the current game situation. Moreover, situations, where a player needs to look carefully for an enemy game character hiding behind the shelter, are widespread. Therefore, reaction and visual search skills seem to be necessary to players in CS:GO. Thus, it is expected that professional players should be superior to novice players in the considered cognitive skills. In addition, some of the mentioned above researches [28–31] are consistent with these assumptions about the reaction skills, and other studies [41–43] indirectly correspond to the assumptions about the visual search skills. In other words, it is assumed that when compared with novice players, professional players can have better (faster) reaction time during the tests on simple reaction time and better reaction time, perhaps along with better correctness, during a task that requires both a high speed of response and decision-making (tasks on choice reaction time). As for visual search skills, we suppose that professional players should spend less time on visual search and have better search results (e.g., higher correct rate) than novices.

Generalizing the above, two separate types of experiments were conducted during the current research: Cognitive Tests (CT) and Oddball paradigm EEG experiments, which were undergone on separate days by the participants. It is interesting not only to study the results of the experiments separately but also to check the relations between them. The fact that EEG data were recorded not during the cognitive tests but during separate EEG experiments means that there is no direct link between EEG signals and the results of the tests on cognitive skills. However, since we assume that the cognitive tests are the physical (functional) representation of brain activity, we can study the relations between the average values of the cognitive test results and EEG features. It is worth noting that, according to the review [6], greater amplitude and shorter latency of P200 [8], N200 [44, 45], and P300 [44, 46] ERP components can be associated to higher capacities inherent in athletes. Reaction time and visual search skills can be part of these higher capacities. Therefore, the mentioned components are of the greatest interest.

Another important question is whether we can use cognitive tests and EEG signals not only to distinguish professional players from novices but to detect the level of the players. Therefore, we also investigated the consistency of the hours spent in CS:GO to the cognitive test results and features extracted from EEG data.

We summarize the following main hypotheses tested in the current study:

1. There exist differences in brain activities of professional and novice players in CS:GO. The differences can be detected in the spectral and temporal domains of the neural signals.

2. The difference in ERP can be related to higher ERP amplitude (due to higher engagement of professional players in the CS:GO game) and shorter latency of ERP peaks, like P300 or N200 (due to faster reaction of professional players). These assumptions were made based on the studies related to the sports domain [4, 6].

3. There can be the superiority of professional players in CS:GO to novices in reaction time, choice reaction time, and visual search.

4. Reaction time, choice reaction time, and visual search can be expressed in the participant's brain activity. In other words, there can be relations between the results of the cognitive tests and EEG experiments.

5. Cognitive tests and ERPs can be used to detect the level of the CS:GO player (based on hours spent in the game). For instance, there can be a correspondence between hours spent in the game and features extracted from cognitive tests and ERP signals.

To emphasize the novelty of our research, we compared similar studies related to EEG and Cognitive tests in Tables S1.1 in (see S1 Appendix) and S2.1 (see S2 Appendix), respectively. The tables show that differences between professional and novice players in FPS games have not been studied based on ERP analysis and cognitive tests. The closest work [37] explores players in a MOBA game (not an FPS game) based on another type of EEG analysis—spectrum analyses of brain activity during playing the game, not ERP analysis of brain responses on key events of the game. As for the impact of the current research, the used methods in the present study are similar to ones applied for studying professional athletes. Hence, the current study can contribute to understanding the influence of eSport on a player's cognitive state and associating eSport with a real sport. In addition, the results of the present study can be beneficial for eSports organizations and professional eSport teams. A more profound understanding of the influence of the video game CS:GO and the importance of several cognitive skills (reaction time, choice reaction time, and visual search) can benefit during recruiting and evaluating team members, as well as making training plans.

## Materials and methods

### Participants

From overall 24 participants (21 male, 3 female), two groups of 10 subjects each were selected for further analysis. The first group was named professional players (PRO). This group consisted of 10 members of the ITMO Esport team Kronbars (Saint Petersburg, Russia, [15]), who participate in eSport competitions in CS:GO discipline (mean age 21 years, ranging from 20 to 22 years). Five of the professional players have from 1800 to 5000 hours spent in CS:GO, and the other five have 5000–8000 hours of experience in the game. It should be noted that all of the participants in the PRO group are active players in CS:GO. Their last games in CS:GO was less than a week before the experiments. The second group was named novice players (NOVICE). The main selection criteria were not participating in eSport competitions (in any discipline) and having less than 250 hours spent in CS:GO. In total, the ten most appropriate participants were selected (mean age 25.3 years, ranging from 22 to 34 years). Eight of them were students of Skoltech University (Moscow, Russia), and two were from MIPT (Moscow, Russia). Six of the members of the NOVICE group were just familiar with the game, one was not familiar, and three were players in CS:GO, but they had less than 250 hours spent in the game. It should be noted that the experiments were conducted in accordance with the Declaration of Helsinki and approved by the local Ethics Committee of Skoltech University. Informed consent was obtained from all participants.

### Experiments procedure

The participants were asked to take part in two types of experiments: EEG experiments and Cognitive Tests. These experiments were separated from each other, and each subject participated only in one type of the experiments per day in order to exclude the possible influence of fatigue.

The EEG experiments consisted of four blocks based on a standard oddball paradigm. The first block had a non-specific Blue/Red Ball (B/RB) paradigm. Non-target and target stimuli were blue and red balls, respectively. All balls (target and non-target) were displayed at the center of a screen and had the same sizes. The next three blocks include game-specific (CS:GO-

related) paradigms: the non-target stimuli image was a screenshot of a game territory without an enemy, and the target stimuli image was a screenshot with an enemy (only one). The pictures of screenshots were at the center of the screen and had the same size, however, the position of the enemy (target) was not fixed. The paradigm names are Big Terrors (BT), Medium Terrors (MT), and Small Terrors (ST), which correspond to the size of the enemy presented on an image. The game-specific blocks were presented in decreasing order of the size of the enemy on the target stimuli: BT, MT, ST. During the Big Terrors paradigm, an enemy is located at the center of the image and can be easily detected. This paradigm is, in essence, similar to the Blue/Red Ball paradigm but related to CS:GO. During the Medium Terrors and Small Terrors paradigms, an enemy can be on different parts of the image, and it is necessary to move the eyes and search for the enemy. Moreover, during the ST paradigm, an enemy can hide behind cover, so only a part of him could be visible (the head, up part of the body, shoulder with the head, etc.). Examples of target and non-target stimuli can be seen in Fig 1.

During the experiments, the participants comfortably sat in front of the screen and looked at the visual stimuli. In the first 30 seconds, a white cross was displayed in the center of the screen, then the random sequence of target and non-target stimuli images were shown. To ensure the participants' attention to the visual stimuli, they were instructed to count the number of target stimuli for each paradigm. After the end of the experimental paradigm, the participants were asked about the number of counted target stimuli images and given a 2–3 minute rest before the next paradigm. Such parameters of the experiments paradigms used in the current study, like stimulus show time (Stimulus duration), time between stimuli (Blank duration), number of target stimuli (# Targets), and number of non-target stimuli (# Non-targets) are presented in Table 1.

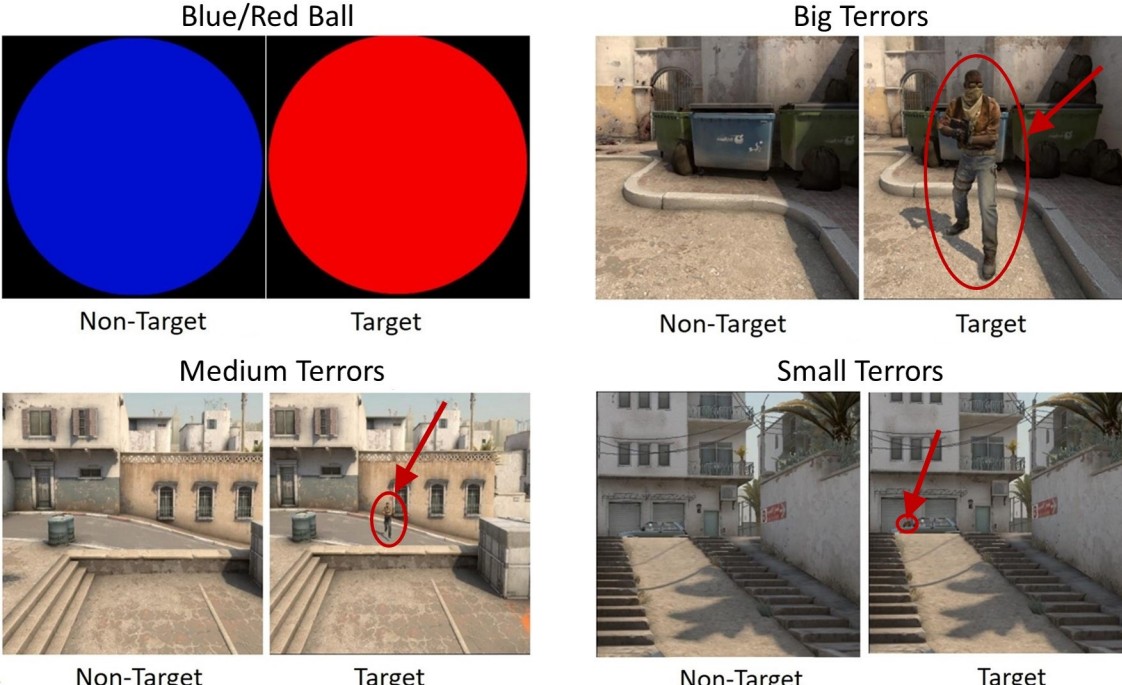

**Fig 1. Paradigms of EEG experiments.** Blue/Red Ball (left top); Big Terrors (right top); Medium Terrors (left down); Small Terrors (right down). The red circles and arrows were added for the convenience of the reader. The stimulus did not contain them.

**Table 1. Main parameters of the EEG experiment paradigms.**

| Block | Stimulus duration (s) | Blank duration (s) | # Targets | # Non-targets |
|---|---|---|---|---|
| B/RB | 0.25 | 0.5 | 60 | 240 |
| BT | 0.4 | 1 | 40 | 260 |
| MT | 0.4 | 1 | 40 | 260 |
| ST | 0.8 | 1 | 56 | 168 |

Cognitive Tests consisted of 4 cognitive tests. The first two tests were on simple reaction time: a test on reaction time using a mouse click as a response action (Reaction Time Mouse) and a test on reaction time using a keystroke as a response action (Reaction Time Keyboard). The third test was on both reaction time and decision-making (Reaction Time Decision), which is essentially a choice reaction time test for our case. Finally, a Visual Search test (VS) was used to estimate the participants' visual search skills. During the cognitive tests, the participants comfortably sat in front of the screen and used an ergonomic computer mouse, mouse pad, and keyboard. Each test lasted not more than 3–4 minutes, and there was a one-minute break between the tests. All cognitive tests were non-specific and had no direct link to CS:GO game. A detailed description of the cognitive tests is presented in S3 Appendix.

## Equipment

The g.Nautilus wearable EEG headset with 32 channels and 250 Hz sampling rate designed by g.Tech was used for recording EEG data. The EEG data were collected using g.HISYS high-speed online processing software for Simulink (Matlab). A special photo device based on Arduino Nano and a photoresistor was designed to provide accurate synchronization of visual stimuli and EEG data. The paradigm presenter of EEG experiments was designed based on Pro Lab software (Tobii Pro). The presenter provides good time stability of stimuli presenting (time fluctuations are less than 5 ms) in combination with a monitor with a high frame rate (240 Hz).

Cognitive Tests are computer tests programmed based on Psychophysics Toolbox 3.0 lib in Matlab [47, 48], which include collecting data from the mouse and keyboard. By default, an ergonomic game mouse, mouse pad, and keyboard were offered for use to all participants, however, each participant could bring and use their habitual devices. In practice, all professional players brought their own items (mouse, keyboard, and mouse pad), while novices did not.

## EEG experiments

**EEG recording.** Recording of the electroencephalogram (EEG) was done by 32 silver/silver chloride electrodes attached to an electrode cap (g.Nautilus) placed according to the extended international 10–20 system. Impedances were kept below 20 kΩ. The ground electrode was located on the upper forehead, and the reference electrode was on the right ear. Preprocessing was done using Python 3.6 and MNE (v.0.23.0).

**EEG preprocessing.** We filtered the continuous EEG with a high-pass 0.5 Hz Kaiser-windowed (beta 5.65) sinc FIR filter (order 9,056) and a low-pass 30 Hz Kaiser-windowed (beta 5.65) sinc FIR filter (order 184). We excluded the first two trials in each block and epochs with amplitude changes exceeding 500 $\mu$V at any channel to exclude non-stereotypical artifacts that are not likely to reflect eye movements or blinks.

In order to remove eye movements, blinks, and Electrocardiogram (ECG) artifacts, ICA was used. Prior to the ICA application, a copy of the EEG signal was created and filtered using

a high-pass IIR filter with a cutoff frequency of 1 Hz (to avoid low frequency domination among components) and further applied to the original signal filtered at 0.5 Hz. We applied a semi-automatic Infomax algorithm for ICA [49] to detect eye movement and blink components via correlation with frontal electrodes (Fp1 and Fp2). The components with higher correlation were further visually examined and discarded to ensure proper blinks removal. ECG-related ICs were selected via visual inspection of time evolution (clear QRS complex), topography (near linear gradient—due to far dipole localization), and spectrum (most power lies below 1–2 Hz and has no spikes) [50, 51]. We identified and interpolated noisy channels and performed automatic rejections of bad epochs using the technique by [52] implemented in the autoreject library [53]. We interpolated EEG channels with unusually high deviations in activity (calculated as a z score exceeding 3.0 with SD of 0.7413 times the interquartile range). The interpolation affected no more than two electrodes per participant in all participants. The epochs rejection discarded 16% of epochs (average) but not more than 30% for a single subject.

**ERPs and spectrograms.**   For paradigms that use pictures from CS:GO game—Big Terrors, Medium Terrors, and Small Terrors, epochs were 1400 ms long, including a 500 ms pre-stimulus. For the simple oddball Blue/Red Ball paradigm, in turn, epochs were 800 ms long, including a 200 ms pre-stimulus. For each recording, we subtracted 100 ms pre-stimulus baseline. ERPs for each subject were calculated as an average of recordings when the target stimulus was presented on the screen. Grand average ERPs were calculated as an average of ERPs for each subject in the corresponding group. Then, the statistical comparison based on Student's t-test with Threshold Free Cluster Enhancement (TFCE) [54] was used for calculating significant differences between the ERPs curves.

In order to avoid distortions caused by the signal averaging process on the ERP components (P200, N200, and P300)calculation, we applied the further technique: The initial latency for ERP components was found by manually expecting each participant's average ERP for each channel. The peak was considered valid if it was in the vicinity of the corresponding component timing. With the increased complexity of a task, the P300 component can be no longer found exactly near 300 ms after stimulus presentation—the fact that we saw in the ST block of EEG experiments. The amplitude of each component was then smoothed as a mean around the selected peak (+/- 10 ms). The peak latency was selected as the point that divides the area under the peak into two halves [55, 56].

Hilbert Spectrograms were computed using the Hilbert-Huang transform [57] of the intrinsic mode functions (IMFs) extracted using variational mode decomposition (VMD) [58]. No IMFs were discarded in order to preserve the whole bandwidth of the EEG signal. We preferred VMD to Wavelet transform because of VMD's better time/frequency resolution [57]. Moreover, it has been demonstrated that IMFs obtained by EMD (and consequently, VMD) provide frequency responses similar to that of a dyadic filter bank [59]. With appropriate initialization, VMD converges to Wavelet Packet Decomposition or STFT-like expansion with time-varying window width [60]. Thus we selected VMD as a more informative decomposition over the alternatives. The frequency band of the spectrograms was selected from 0 to 30 Hz. After computing spectrograms for each trial of each participant in both groups, we calculated the mean spectrogram of target stimuli and mean spectrogram of non-target stimuli. These mean spectrograms were used in statistical comparisons, which were proceeded with applying Student's t-test and TFCE.

**Tensor decomposition.**   Although ERP and Spectrograms give us a good estimation of average brain dynamics, it would be interesting to figure out specific "main" patterns of the EEG signals that are different or similar among the groups. Common methods like ICA or PCA cannot take into consideration the multidimensionality of the data, while the tensor

decomposition can do it [61] and has been successfully applied to EEG signals [62]. We performed two types of analyses based on Canonical Polyadic Decomposition (CPD): Spatial-Temporal and Spatial-Temporal-Spectral, accordingly, for ERPs and Spectrogram data.

For Spatial-Temporal analysis, we considered order-3 tensors formed of channels (32), time (251), and trials (670–1000). From the ERPs after preprocessing, we subtracted the offset based on the first 80 ms of each trial (epoch). We considered the 100 ms pre-stimulus part and the full part after the stimulus.

The tensor for Spatial-Temporal-Spectral analysis was constructed by concatenation of the Hilbert spectrograms computed in the previous section for each trial and channel. That leaves us an order-4 tensor of channels(32), frequency (101), time (151), and trials (670–1000). There are 101 frequency points from 0 to 30 Hz. The spectrograms were also cropped from 100 ms before and 500 ms after the stimulus since beyond these limits the spectrograms can be distorted by the boundary effects.

The CPD of the space-time-trial tensor gives 3 factor matrices composing spatial and temporal components and trial intensity. The analogous is for the space-time-frequency-trial tensor that forms 4 factor matrices composing spatial, temporal, and spectral components and trial intensity. In order to recover "power" that a specific CPD component has for each subject, we summed up the coefficients related to the subject in the decomposition. Then, we applied Mann-Whitney u-test [63] to calculate the statistical differences between the PRO and NOVICE groups. The SciPy Python library was used to calculate the statistical corporation.

We applied CP decomposition with orthogonality constraints on spatial to the components [61, 64], to prevent the degeneracy of CPD components. All CP decompositions of the data were performed using Matlab.

## Cognitive tests

In addition to the cognitive tests themselves, we used the counted number of target stimuli during EEG experiments as a marker of cognitive abilities. During MT and ST paradigms, the terrors could appear at a different part of the picture, therefore, the count error of the target stimuli can characterize visual search skills. It should be emphasized that counting errors were used the same way as all other metrics of the cognitive tests, accordingly, the corresponding sections related to Cognitive Tests also mention these paradigms of the EEG experiments.

**Data preprocessing and analysis.** The data from each cognitive test were used to calculate the result parameters (or metrics). All parameters for professional players were aggregated into PRO group metrics, and all parameters for novice players were aggregated into NOVICE group metrics. These metrics aggregated into groups (hereinafter group metrics) are formed as the following. After filtering (more details below) the data for each test subject separately, all trials for all test subjects belonging to the same group were merged in the single sample as an overall population. These group metrics were used for statistical analysis. It should be noted that if it is a complex parameter calculated for the test subject from all test trials (like correct rate, final score, etc.), the parameter itself was aggregated into the group metric.

For example, let us assume that the reaction time test with decision-making (choice reaction time) consisted of 30 trials and was conducted for 10 members of a group. It means that there are 30 values of reaction times for each of 10 members. However, there are only 10 values of correct rates, as correct rate parameters are calculated based on all subject's trials. Thus, in this situation, the group metric "Reaction time" will consist of 300 values (if filtering is not taken into account), which can be used for evaluating the overall group. The group metric "Correct rate", in turn, will consist of 10 values.

Filtering of the data was applied for each subject separately and consisted of dropping out unrealistic and inadequate values (like reaction time less than 100 ms or more than 1 s), as well as removing outliers (based on 1.5 interquartile range (IQR)), if applicable. A permutation test based on Student's t-test for unequal variance [65–67] was used for group metrics to compare distribution between the groups. This permutation test was chosen as it is more reliable than an ordinary Student's t-test and is good for interpretation (evaluates the differences between mean values). The base p-value threshold for the permutation test was chosen as 5%. The practical realization of the statistical tests was made in Python using SciPy stats lib [68]. To calculate the effect size, Hedges's standardized mean difference g with correction for small sample size was selected [69]. Hedges's *g* is similar to Cohen's *d* but more robust for different sample sizes.

All group metrics used during the corresponding cognitive tests are listed in Table 2. These metrics are explained in detail in the following sections, and a full description of the cognitive tests can be seen in S3 Appendix.

**Metrics of reaction time tests.** The main parameter (metric) for the reaction time tests was reaction time for Reaction Time Mouse (RTM) and Reaction Time Keyboard (RTK) tests and reaction time during correct mouse clicks (correct reaction time) for Reaction Time Decision (RTD) test. For these parameters, firstly, all trials in which the reaction time was less or more than the corresponding thresholds (see Table 3) were excluded. The thresholds were chosen based on the physiological limits of human reaction time and the results of the previous research [70]. Then, the filtration (removing outliers) was conducted separately for each test subject. Finally, the filtrated parameters were aggregated into group metrics. In addition, the correct rate during the RTD test was used as a group metric. The correct rate was computed as the percentage of the correct mouse clicks (left click for the red circle and right click for the blue circle) from all mouse clicks.

**Metrics of visual search test.** For the Visual Search test, the main parameters (metrics) were correct rate (percentage of the correctly solved search tasks from all passed search tasks), final score (initially zero, +1 for a right answer, and -1 for a wrong answer), and the particular time intervals during correct answers: search times of "L" and search times of "no L"

**Table 2. Metrics used for assessment of the cognitive abilities.**

| Experiment | Metrics |
|---|---|
| Reaction Time Mouse (RTM) | Reaction time |
| Reaction Time Keyboard (RTK) | Reaction time |
| Reaction Time Decision (RTD) | Reaction time (correct), Correct rate |
| Visual Search (VS) | Final Score, Search time of L (correct), Search time of no L (correct), Correct rate |
| Medium Terrors (MT) | Count error |
| Small Terrors (ST) | Count error |

**Table 3. Reaction time thresholds for reaction time tests.**

| Experiment | Min RT (ms) | Max RT (ms) |
|---|---:|---:|
| Reaction Time Mouse | 120 | 500 |
| Reaction Time Keyboard | 120 | 500 |
| Reaction Time Decision | 140 | 700 |

(respectively, search time of L and search time of no L). All these parameters were used as group metrics.

**Metrics of cognitive abilities from EEG experiments.** As mentioned above, the count error of the target stimuli can be a parameter (metric) estimating the visual search skills for the Medium Terrors and Small Terrors paradigms of the EEG experiments. Count error was defined as the ratio of the count difference between the calculated value and the correct number to the correct number. Count error was expressed in percentage.

## Relations between cognitive tests and EEG experiments

Cross-decomposition methods like partial least squares (PLS) and canonical correlation analysis (CCA) used in many research [71–75] are not useful in our study as these techniques require a relatively high number of features to subject [76, 77]. Therefore, Spearman correlation analysis was used to determine correlations between Cognitive Tests and ERPs. To provide a reliable p-value level, Threshold Free Cluster Enhancement (TFCE) with permutation test (100000 permutations) was used [78].

We performed a correlation analysis between EEG and CT features separately for the PRO and NOVICE groups (intragroup analysis) to avoid distortion from intergroup differences. The ERP features include mean latency and amplitudes of ERP peaks (P300, N200, P200) during target stimuli for all EEG paradigms (BR/B, BT, MT, ST) and all EEG channels. In addition, the mean values of these features for Fz, Cz, and Pz channels were used as combined ERP features for preliminary estimation of ERP correlating with CT features. Thus, there were 24 combined features, which corresponded to each considered ERP peak (P200, N200, and P300) for each ERP paradigm (B/RB, BT, MT, and ST) for both amplitude and latency. As for Cognitive Tests, the mean values of reaction times in RTM, RTK tests, mean correct reaction time, and correct rate in RTD test, as well as final score, correct rate, mean search times of "L" and "no L" for VS test, were used. All these values were calculated for each participant and were used as CT features.

## Correspondence of a player's level with the CT results and ERP signals

It is obvious that player level could have a positive correlation with the hours spent in the game. Therefore, hours spent in CS:GO for each member of the PRO group were used as a special feature. We calculated the correlation of this feature between this feature and both CT and ERP features by the same method used in the previous subsection.

## Results

### Cognitive tests

The main results of the comparison between professional and novice payers are presented in Table 4. The statistical analysis of cognitive tests related to reaction skills (RTM, RTK, and RTD) showed statistically significant differences in reaction time (reaction time (correct) for RTD). The distributions of these RT metrics are shown in Fig 2. More specifically, the professional players have 52 ms (p≈0) faster reaction time in the RTM test, 25 ms (p≈0) faster RT in the RTK test, and 64 ms (p≈0) less RT in the RTD test.

It is important to note that the considered cognitive tests measure whole reaction time, including the time of the response action. The statistical comparison results showed the superiority of the PRO group for both mouse click and keystroke response actions. The necessity to make a choice (decision-making) during the RTD test also does not change the superiority of the professional players. However, correct rate of the PRO group, in turn, does not

**Table 4. Statistical analysis of cognitive tests results between the professional and novice players.**

| Exp | Group metric | PRO | NOVICE | Test* | Test value | p** | g *** |
|-----|--------------|-----|--------|-------|------------|-----|-------|
| RTM | **Reaction time** | 219±32 ms | 271±61 ms | **t(561)** | **-12.7** | **0.0000** | **1.07** |
| RTK | **Reaction time** | 242±35 ms | 267±33 ms | **t(560)** | **-8.8** | **0.0000** | **0.74** |
| RTD | **Reaction time (correct)** | 332±74 ms | 396±92 ms | **t(515)** | **-8.8** | **0.0000** | **0.77** |
| RTD | Correct rate | 88.3±3.2% | 89.7±7.8% | t(18) | -0.5 | 0.71 | 0.21 |
| VS | Final score | 16.90±6.05 | 19.10±6.37 | t(18) | -0.8 | 0.46 | 0.34 |
| VS | Search time of L (correct) | 2.53±1.73 s | 2.33±1.26 s | t(185) | 0.9 | 0.37 | 0.13 |
| VS | Search time of no L (correct) | 5.16±2.52 s | 5.22±3.74 s | t(226) | -0.2 | 0.88 | 0.02 |
| VS | Correct rate | 87.2±6.7% | 92.0±6.0% | t(18) | -1.7 | 0.11 | 0.73 |
| MT | Count error | -0.3±0.8% | 0±1.7% | t(18) | -0.4 | 1 | 0.18 |
| ST | **Count error** | 0.4±4.8% | -7.3±5.7% | **t(18)** | **3.24** | **0.0064** | **1.39** |

\* Permutation test based on Student's t-test for non-equal variance with 100000 permutations.

\*\* **p < 0.05** are highlighted in bold

\*\*\* *g*—Hedges's standardized mean difference with corrections for small sample sizes

significantly differ from correct rate of the NOVICE group (p = 0.71) as presented in Table 4 and in Fig 2. In other words, there was no superiority of the professional players in decision-making during RTD.

The corresponding distributions (Fig 3) were built to study more about the influence of a response action and a decision-making necessity. The difference in the mean values of reaction time during RTM and RTK was statistically significant only for the professional players. In other words, the reaction time during a mouse click response action (in RTM) is about 23 ms (p≈0) shorter than during a keystroke response action (in RTK) for the PRO group. For the

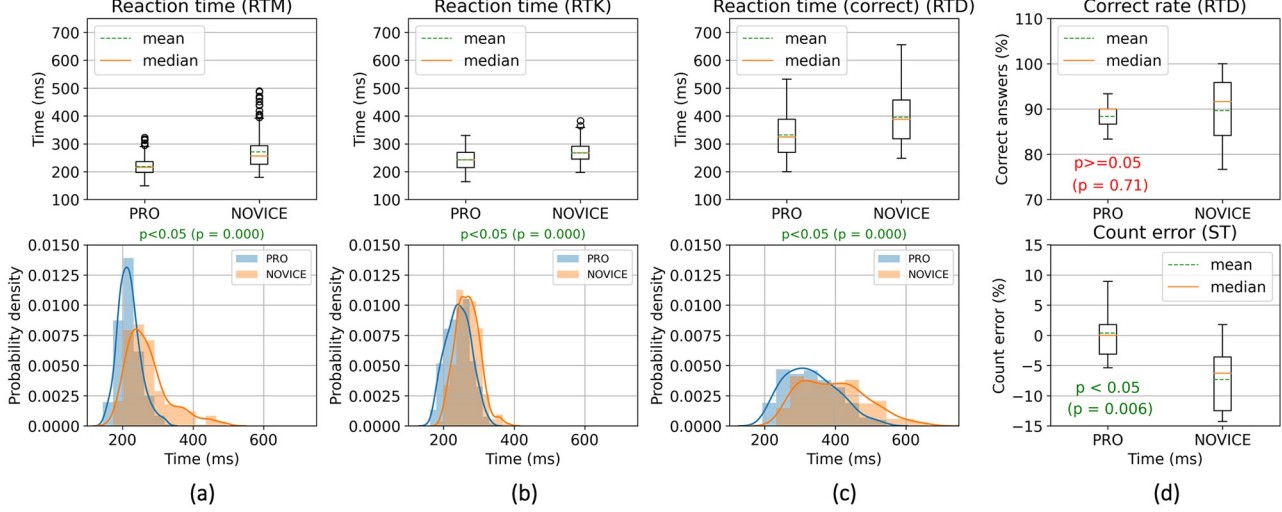

**Fig 2. Statistical comparison between the professional and novice players for cognitive tests results.** Columns: (a) reaction time for RTM test; (b) reaction time for RTK test; (c) reaction time (correct) for RTD test; (d) correct rate for RTD test and count error for Small Terrors (ST) paradigm of EEG Experiments. The comparisons consist of a boxplot (top) and a histogram with a distribution plot (bottom). For column (d), there are only boxplots, but for different group metrics. The used statistical test is a permutation test based on Student's t-test for non-equal variance with 100000 permutations.

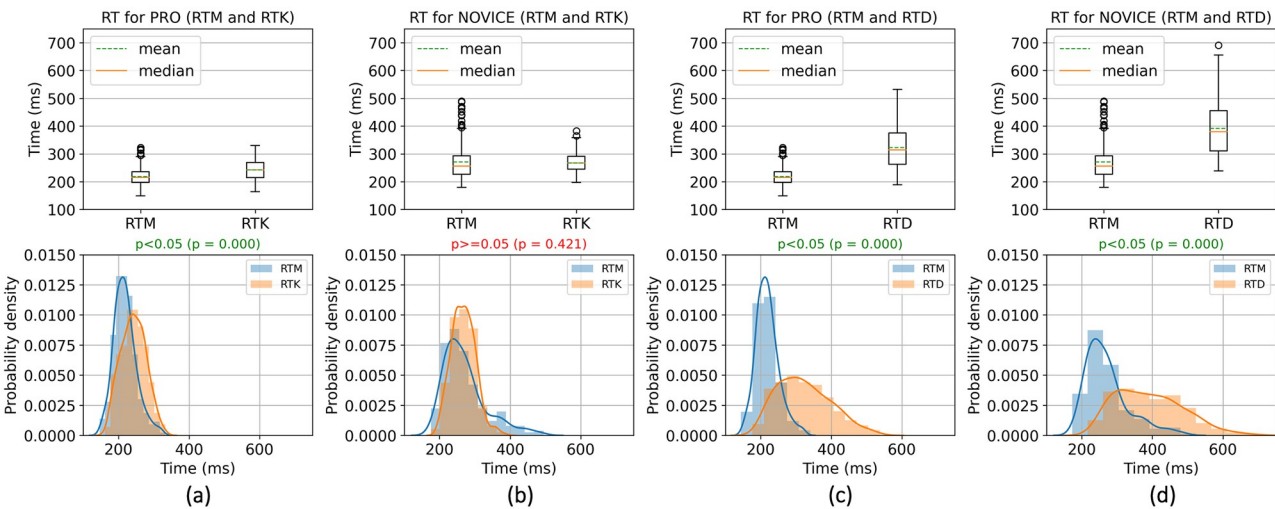

**Fig 3. Statistical comparison of simple and choice reaction time tests for the professional and novice players.** Columns: (a) reaction time for PRO group between RTM and RTK tests; (b) reaction time for Novice group between RTM and RTK tests; (c) reaction time for Pro group between RTM and RTD tests; (d) reaction time for Novice group between RTM and RTD tests. Each comparison consists of a boxplot (top) and a histogram with a distribution plot (bottom). The used statistical test is a permutation test based on Student's t-test for non-equal variance with 100000 permutations.

Novice group, the similar difference is not statistically significant (p = 0.42). As for the necessity of decision-making, the distributions (Fig 3(c) and 3(d)) showed that this necessity increases the reaction time by 113 ms (p≈0) for PRO and 125 ms (p≈0) for NOVICE. These increases are statistically significant.

For testing visual search skills, all statistical comparisons for the VS test did not demonstrate statistically significant differences. In other words, the Visual Search test did not provide any evidence that the PRO group is superior to the NOVICE group in visual search. It should be noted that the VS test estimated general visual search skills that do not relate to CS:GO game directly.

As for the EEG paradigms that required visual search (MT and ST), the mean count error did not exceed 15% in absolute values. During the MT paradigm, most participants counted the correct number of target stimuli, leading to approximately the same distribution of the count error for both groups (p≈1). However, for the Small Terrors, count error are significantly different (see Table 4) between the groups. The professional players have 6.9% (p = 0.006) smaller count error than the novice players. In other words, the PRO group showed better results in the visual search task directly related to CS:GO game. A boxplot of count error during the ST paradigm is presented on Fig 2(d).

## EEG experiments

**ERP.** ERP analysis results are summed up in Fig 4. When comparing grand ERPs between the groups, one can see the component latency being smaller for the PRO group in all experiments. The amplitude, however, has significant differences only in CS:GO-related blocks of EEG experiments (BT, MT, and ST).

In order to compare groups' ERP components quantitatively, we analyzed the latency and peak of each component and each subject. The results are summarized in Table 5. It represents the statistical comparison of latency and amplitude of ERP peaks between PRO and NOVICE groups. It can be seen that, for the professional players, the game-specific paradigms (BT, MT,

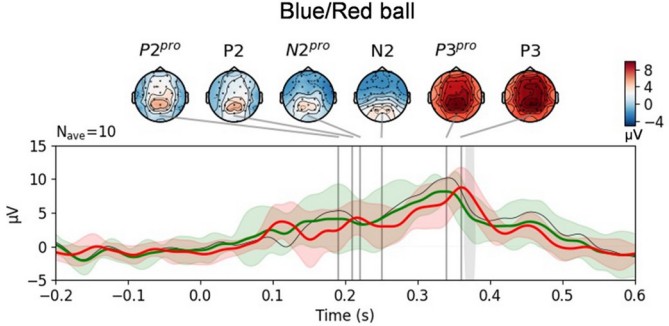

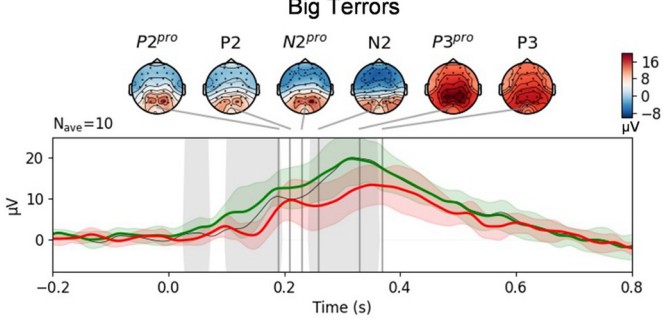

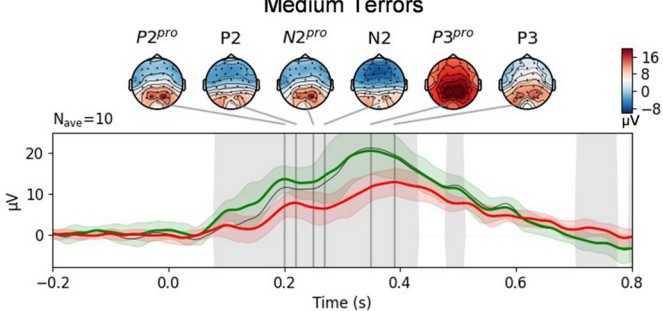

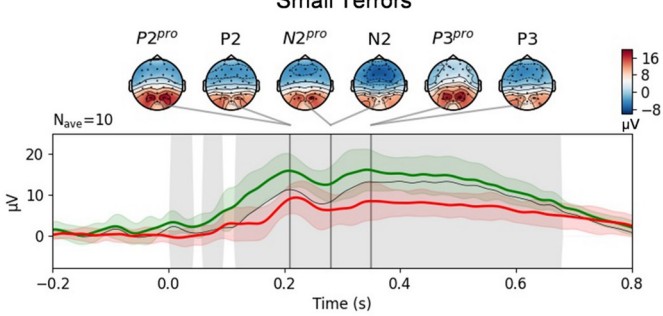

**Fig 4. ERP comparison of PRO and NOVICE groups.** Each plot corresponds to an experiment paradigm. ERPs are calculated as grand average over P3, Pz, P4, PO3, and PO4 electrodes; Green line—professional's ERP, red line—novice's ERP (std—colored shadow regions); Shaded area indicates the regions where PRO and NOVICE groups are significantly different (p<0.05). Times for topographic maps correspond to P200 (P2), N200 (N2), and P300 (P3) components. Components times were selected manually after visual inspection (exact times can be found in Table 5). *P300 for ST paradigm is nominal and was selected as the highest amplitude of EEG during the whole stimulus presentation after N2.

**Table 5. Comparison of ERP components.**

| Latency (ms) comparison | | | | | Amplitude ($\mu$V) comparison | | | | |
|---|---|---|---|---|---|---|---|---|---|
| Exp | Comp | PRO | NOVICE | p** | Exp | Comp | PRO | NOVICE | p** |
| B/RB | **P200** | **177±22** | **208±20** | **0.0050** | B/RB | P200 | 6.78±4.08 | 4.84±2.63 | 0.22 |
| | **N200** | **221±19** | **250±20** | **0.0050** | | N200 | 1.35±4.57 | 1.42±2.79 | 0.97 |
| | **P300** | **335±29** | **361±13** | **0.0094** | | P300 | 10.66±2.83 | 9.28±3.38 | 0.33 |
| BT | **P200** | **187±15** | **209±14** | **0.0034** | BT | P200 | 13.64±4.84 | 10.42±5.19 | 0.17 |
| | **N200** | **228±27** | **261±20** | **0.0080** | | N200 | 11.22±5.41 | 6.65±5.78 | 0.084 |
| | **P300** | **324±25** | **362±22** | **0.0028** | | **P300** | **21.44±6.78** | **14.68±5.49** | **0.0242** |
| MT | **P200** | **195±12** | **217±14** | **0.0017** | MT | **P200** | **14.29±2.97** | **8.41±3.24** | **0.0007** |
| | **N200** | **245±34** | **278±24** | **0.0258** | | **N200** | **10.04±4.05** | **5.22±3.21** | **0.0100** |
| | **P300** | **347±25** | **389±17** | **0.0004** | | **P300** | **21.84±6.11** | **13.48±3.54** | **0.0016** |
| ST | P200 | 209±22 | 222±15 | 0.14 | ST | **P200** | **17.30±3.78** | **9.91±4.44** | **0.0011** |
| | N200 | 273±31 | 289±20 | 0.17 | | **N200** | **10.90±3.82** | **4.69±4.23** | **0.0037** |
| | **P300** | **493±52** | **568±32** | **0.0003** | | **P300** | **15.87±5.40** | **7.75±3.16** | **0.0006** |

*—P300 for ST was determined as maximal potential achieved after stimulus presentations and does not represent standard ERP component but rather a maximal reaction (time and amplitude) during visual search task

**—Permutation test based on Student's t-test for not equal variance. During calculations of the values, the test was performed with 100000 permutations.

and ST) elicit much stronger ERPs (greater amplitude) of P300 (by 8.12 $\mu$V (p = 0.0006) for ST, 8.36 $\mu$V (p = 0.0037) for MT, 6.72 $\mu$V (p = 0.0242) for BT), of N200 (by 6.21 $\mu$V (p = 0.0037) for ST, 4.82 $\mu$V (p = 0.01) for MT), and P200 (by 7.39 $\mu$V (p = 0.0011) for ST, 5.88 $\mu$V (p = 0.0007) for MT, 3.03 $\mu$V (p = 0.17) for BT). In other words, only the differences in amplitude of P200 and N200 for the BT paradigm were not statistically significant for CS:GO-related paradigms. As for the simple B/RB paradigm, there are no significant differences in the amplitudes of the ERP peaks. The statistical comparison of latency of the ERP peaks revealed that the PRO group showed faster P300 for all paradigms (by 75 ms (p = 0.0003) for ST, 42 ms (p = 0.0004) for MT, 38 ms (p = 0.0028) for BT, and 27 ms (p = 0.0094) for B/RB). In addition, N200 was faster by 33 ms (p = 0.0258) for MT, 33 ms (p = 0.008) for BT as well as by 29 ms (p = 0.005) for B/RB. A statistically significant difference in P200 latency (shorter for the PRO group) was also observed for BR/B (22 ms with p = 0.005), BT (31 ms with p = 0.0034), and MT (22 ms with p = 0.0017). Thus, it is possible to summarise that the appearance timing of almost all components of professionals is on average 20–70ms earlier than similar in the NOVICE group (see S5 Appendix in addition to Table 5).

**Spectrograms.** EEG spectral dynamics [79] results across four paradigms are presented as a comparison of EMD of Cz channel data for the Pro vs Novice group, enhanced by TFCE in Fig 5. In this figure, the white zones represent statistically non-significant differences, and red-colored and blue-colored—statistically significant positive and negative differences, respectively.

One can easily spot that PRO has stronger stimulus-locked alpha-band power in game-related blocks of EEG experiments. It also should be noted that the professional players had slightly stronger beta power in the occipital electrodes than the novices and statistically significant beta suppression in the right frontal electrodes (see S4 Appendix).

**Spatial-temporal analysis.** Order-3 (subjects-channel-time) tensors of EEG signals for four experimental paradigms were approximated by the CP decompositions with rank-5, which provided 5 sets of the associated subject, spatial, and temporal loading components. Among the results, the two most relevant components with the highest consistency across

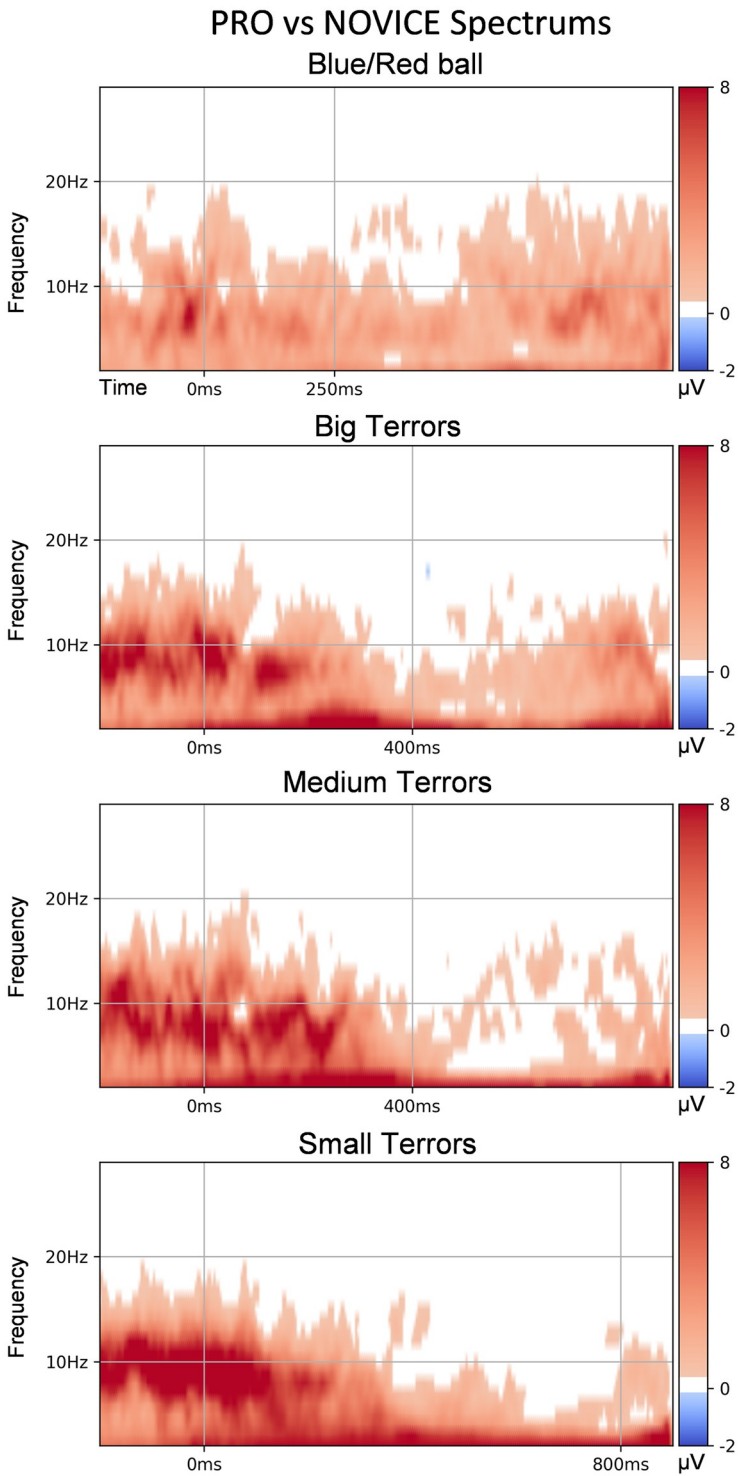

**Fig 5. Spectrograms comparison of PRO and NOVICE groups.** Each row corresponds to a separate experiment paradigm (from top to bottom)—BR/B, BT, MT, ST. ERP spectrograms are calculated on the Cz electrode; Statistically non-significant regions ($p > 0.05$ for nonparametric TFCE) are colored white. Significant regions ($p < 0.05$) are colored as the difference in mean spectrograms for the Pro reaction to the target stimulus versus the Novice reaction to the target stimulus. To not show the spectrogram distorted by the boundary effects, the presented time range was limited by 100 ms before and 500 ms after the stimulus.

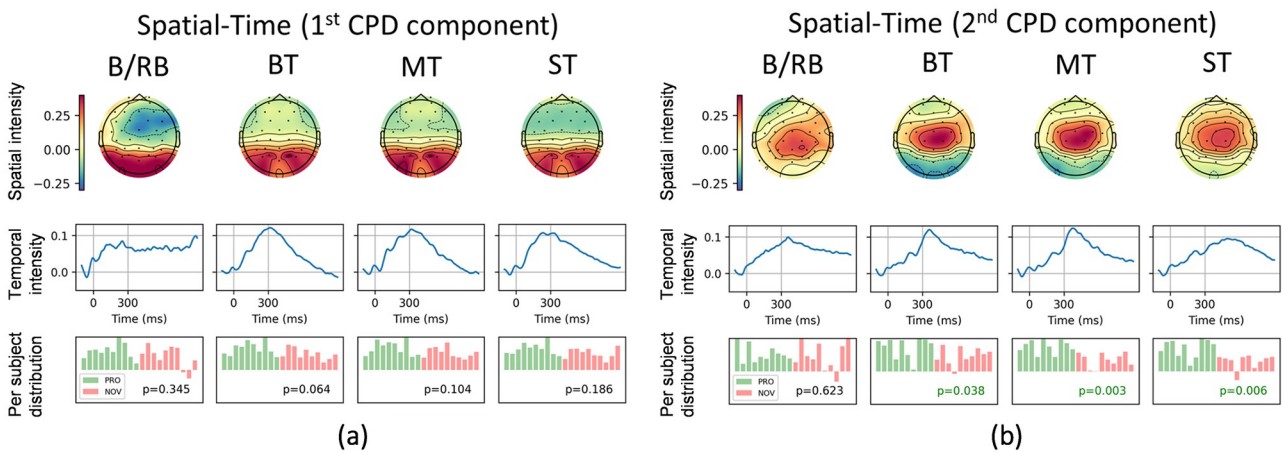

**Fig 6. First\* 2 components of CP decomposition of spatial-temporal tensor for different EEG paradigms.** Left block—first component, right block —second component. Paradigms: Blue/Red Ball, Big Terrors, Medium Terrors, and Small Terrors. For each paradigm, the rows represent the following factors (from top to bottom): electrode (spatial) topographic map, time graph, and per-subject distribution (professionals—green, novices—red) of component power. For each component and row, the range of the vertical axis (or colorbar) is made the same \*Due to the permutation ambiguity of CPD, the components were realigned.

experiments in spatial and spectral domains are presented in Fig 6. The first row represents the spatial distribution of component power (channels), the second row—time evolution of the component, and the last row—distribution of component power among participants in two groups (green—PRO, red—NOVICE) formed from the trials dimension.

For the CS:GO-related paradigms (BT, MT, and ST), the presented components are similar. The first CPD component has a high positive intensity in the occipital cortex (PO3, PO4, PO7, and PO8 channels). The allocation of the second component is in the central part of the head (Pz, Cz, CP1, and CP2). The temporal evolution of both components is shaped like an ERP signal with distinguishable peaks such as P200, N200, and P300. The per-subject distribution shows statistically significant differences ($p < 0.05$) between the PRO and NOVICE groups for the second CPD components for all game-specific paradigms. The p-values are 0.038, 0.003, and 0.006 for BT, MT, and ST, respectively.

As for the Blue/Red Ball paradigm, the CPD component has similar spatial and temporal intensity distributions as the ones for the game-specific paradigms. However, the first component has some differences in spatial and temporal dimensions. The Pz channel has more power, and the temporal intensity does not decrease after 300 ms. The per-subject distributions, in turn, do not significantly differ for both CPD components related to the B/RB.

**Spatial-temporal-spectral analysis.** The results of the spatial-temporal-spectral analysis are presented in Fig 7. For each CPD component, the first row represents the spatial distribution of component power (channels), the second row—time evolution of the component, the third row—frequency distribution, and the last row—distribution of component power among participants in two groups (green—PRO, red—NOVICE) formed from the trials dimension. It is worth noting that the first and second CPD components were extracted from the 4-dimensional space-time-frequency-trial tensor with a frequency band from 0 to 30 Hz. The decomposition results showed the extremely high intensity of low-frequency components, which correspond to ERPs. Because of this, the CPD decomposition was also performed for a high-frequency band from 3 to 30 Hz. The first component of this decomposition is presented in Fig 8, which represents alpha waves (high intensity between 8 and 12 Hz).

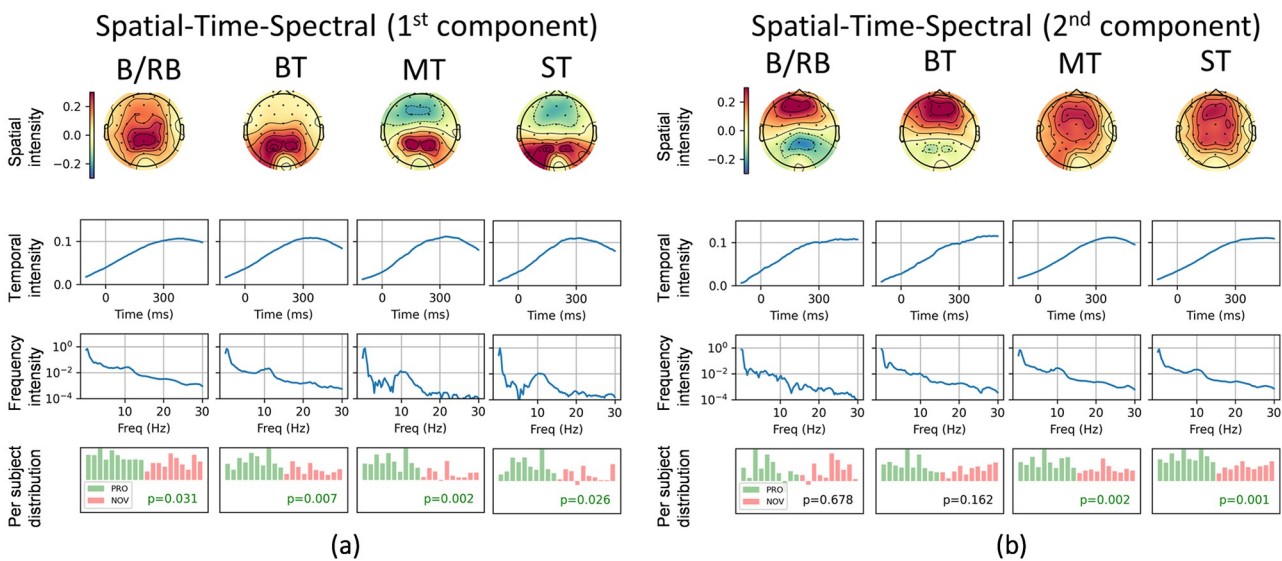

**Fig 7. Results of spatial-temporal-spectral analysis.** Paradigms: Blue/Red Ball, Big Terrors, Medium Terrors, and Small Terrors. For each paradigm, the rows represent the following factors (from top to bottom): electrode (spatial) topographic map, time graph, spectral (frequency) graph, and per-subject distribution (professionals—green, novices—red) of component power.

The CP decomposition of the time-frequency-trial tensor provided two meaningful components (Fig 7). The first component is allocated in the occipital region (Pz, PO3, and PO4). The temporal intensity grows up to about 300 ms and then decreases. The spectral distribution has a decreasing shape with a small local peak for alpha waves. The statistical test on per-subject distribution showed significant differences for all EEG paradigms. The p-values are 0.031, 0.007, 0.002, and 0.026 for B/RB, BT, MT, and ST, respectively.

The second CPD components correspond to the frontal cortex (Fz, F3, F4, FC2, FC2) for all paradigms. For MT and ST paradigms, the intensity in the parietal cortex (Pz, CP1, CP2, and Cz) is also high. The time distributions increase with the peak point after 300 ms. The spectral intensities of the second CPD components decline to high frequency. One can observe the small peaks of about 10 Hz in the spectrum for MT and ST. The pre-subjects distributions for these paradigms show significant differences with p = 0.002 for MT and p = 0.001 for ST.

As for spatial-temporal-spectral analysis for the high-frequency band (from 3 to 30 Hz, without low frequencies), the meaningful CPD component can be related to alpha waves (Fig 8). For all EEG paradigms, the component is allocated in the occipital area with peaks in PO3 and PO4 channels. The temporal intensity decrease after the stimulus. The spectrum has a pronounced peak close to 10 Hz (alpha waves). However, the components do not show significant differences according to their per-subjects distribution.

## Relations between cognitive tests and EEG experiments

To get a broad picture of the correlations between the considered features, the correlations between all CT features, hours spent in the game (for the PRO group), and combined EEG features were separately calculated for both PRO and NOVICE groups. The corresponding correlation heatmaps are presented in Figs 9 and 10.

**Correlation between EEG experiment paradigms.** For the PRO group, combined ERP features of mean amplitude for CS:GO-related paradigms have a relatively high and positive

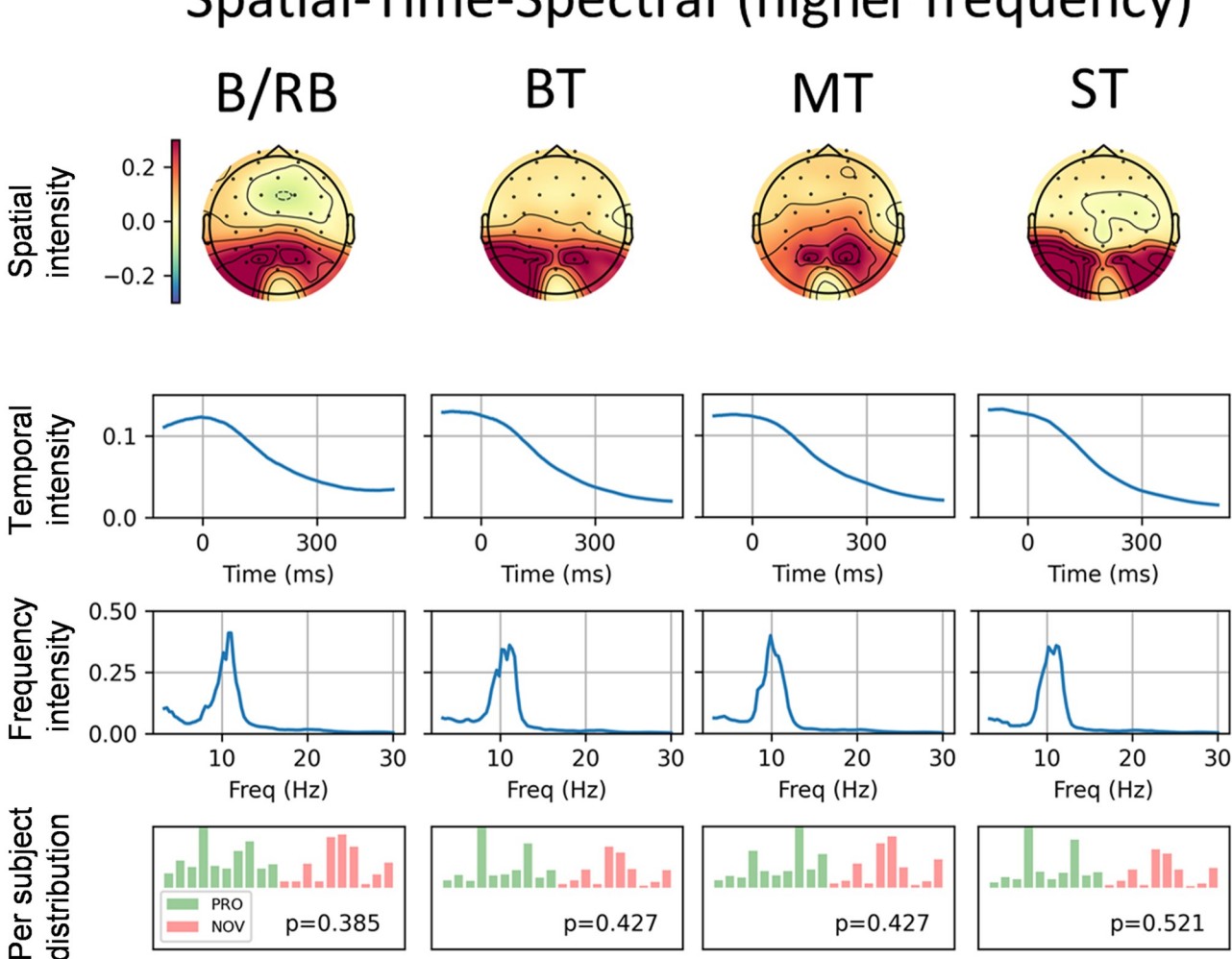

**Fig 8. Results of spatial-temporal-spectral analysis for higher-frequency band\*.** Paradigms: Blue/Red Ball, Big Terrors, Medium Terrors, and Small Terrors. For each paradigm, the rows represent the following factors (from top to bottom): electrode (spatial) topographic map, time graph, spectral (frequency) graph, and per-subject distribution (professionals—green, novices—red) of component power. * high-frequency band is from 3 to 30 Hz (without low frequencies).

correlation with each other, as shown in Fig 9. However, the correlations between these paradigms and Blue/Red Ball are lower. A similar picture, but to a lesser extent, is observed for combined ERP features related to the mean latency. In other words, the CS:GO-related paradigms are distinguished from the non-specific paradigm (B/RB) for professional players. For the NOVICE group, in turn, the correlations between all paradigms are more uniform (see Fig 10). In addition, the negative correlations between the mean amplitude of P300 and mean latencies of P200, N200, and P300 for CS:GO-related experiments are observed for the PRO group. It is especially visual for medium and Small Terrors paradigms, where correlations are statistically significant. The results of the NOVICE group did not show a similar allocation of correlation.

**Correlation between cognitive tests.** No high and significant correlation is observed between RT and VS features for both groups. However, there are relatively high positive

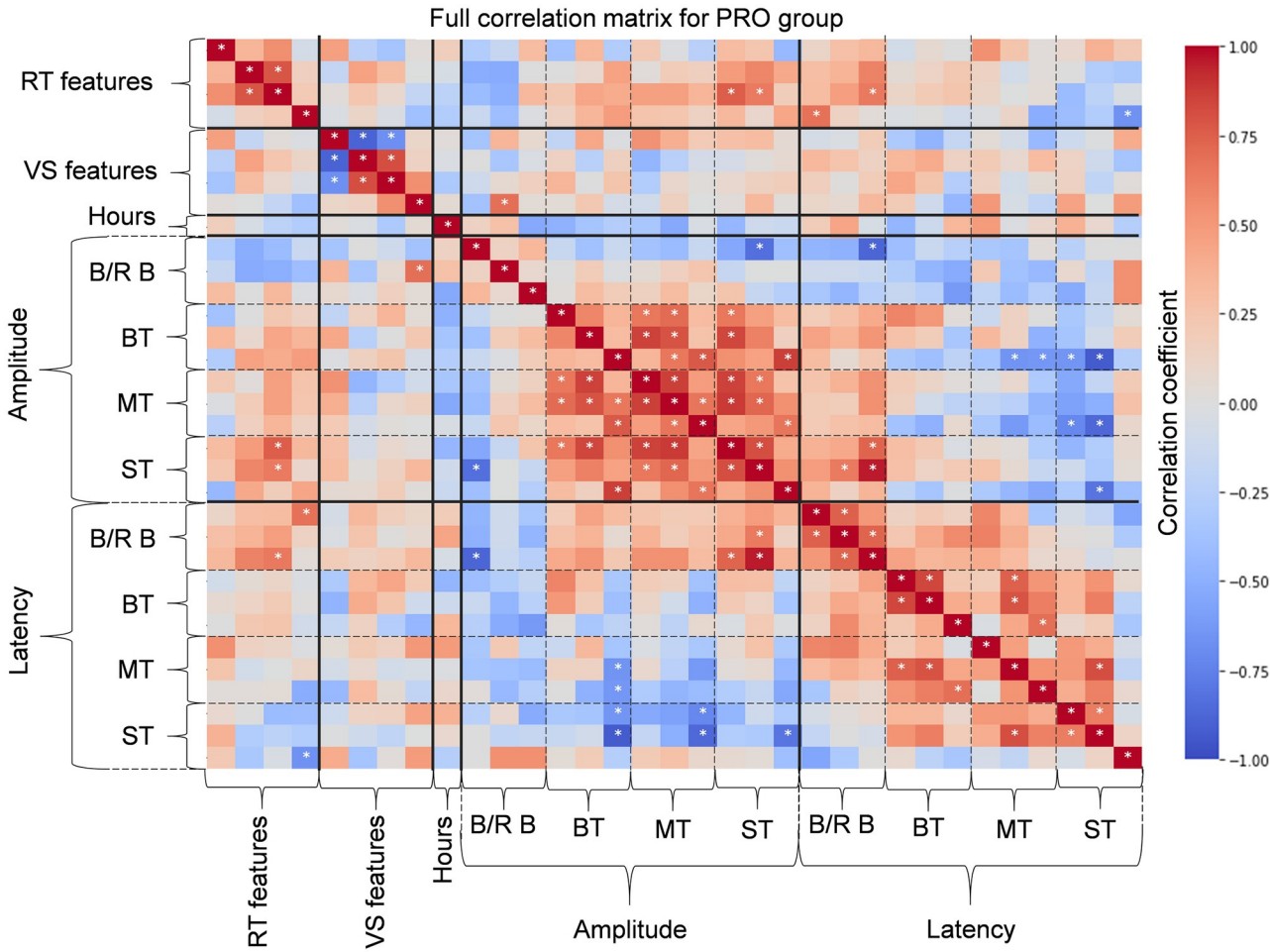

**Fig 9. Correlation heatmap for PRO group.** All features are in the following order. RT features: mean reaction times in RTM and RTK tests, mean correct reaction time during correct clicks and correct rate in RTD test; VS features: final score, mean search time of "L", mean search time of "no L", and correct rate; Hours spent in CS:GO; mean amplitudes of P200, N200, and P300 peaks during target stimuli for B/RB, BT, MT, ST EEG paradigms; mean latency time of P200, N200, and P300 peaks during target stimuli for B/RB, BT, MT, ST EEG paradigms. Correlations with p-value < 0.05 are marked with the symbol "*".

correlations between mean reaction times for all RT tests (RTM, RTC, and RTD) for the PRO group. However, only the correlation of 0.782 between the reaction times for RTK and RTD was statistically significant (p = 0.0075), while the correlation of 0.346 between the reaction times for RTM and RTK was non-significant (p = 0.34). For the NOVICE group, a correlation of 0.515 (p = 0.13) is observed between mean reaction times for RTM and RTK tests, which is not significant. As for visual search or VS features, it is worth noting that final score has a significant negative correlation with mean search time of "no L" for both groups. The correlation values are -0.675 (p = 0.032) and -0.942 (p = 0.000045) for the PRO and NOVICE groups, respectively.

**Correlation between CT and EEG features.** Due relatively low number of members in the considered groups (10 members in each), we used several criteria to determine the correlation as significant for the analysis of the correlations between CT and EEG features:

- Correlations should be comparable for close experiments. For instance, MT and ST are required to search the terrors, so their correlations should be similar.

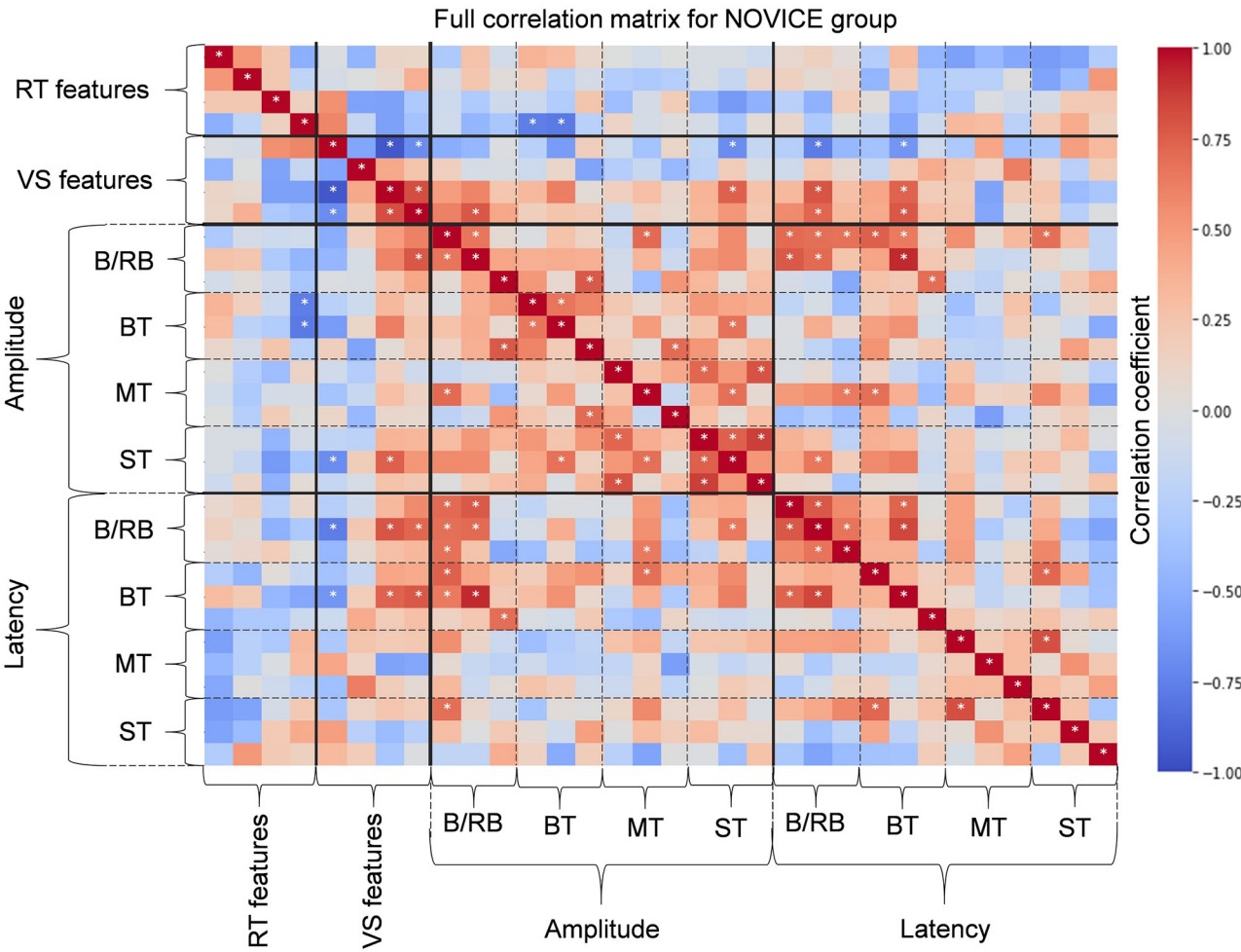

**Fig 10. Correlation heatmap for NOVICE group.** All features are in the following order. RT features: mean reaction times in RTM and RTK tests, mean correct reaction time during correct clicks and correct rate in RTD test; VS features: final score, mean search time of "L", mean search time of "no L", and correct rate; mean amplitudes of P200, N200, and P300 peaks during target stimuli for B/RB, BT, MT, ST EEG paradigms; mean latency time of P200, N200, and P300 peaks during target stimuli for B/RB, BT, MT, ST EEG paradigms. Correlations with p-value < 0.05 are marked with the symbol "*".

- There should be several statistically significant channels after TFCE with permutation test in the area of the correlations.

During the analysis, we found only two CT features that satisfy these two criteria. The first is correct rate in RTD, which has a high negative correlation in the occipital area for the participants of the NOVICE group. Most channels with significant correlation are observed predominantly for P200 and N200 during CS:GO-related paradigms (ST, MT, and BT) in the occipital cortex (Fig 11). Besides, significantly correlated channels are also located in the parietal cortex for the BT paradigm. It is worth noting that a similar allocation of the correlations can be found for almost all EEG paradigms and ERP components. For the PRO group, in turn, the correlation between correct rate in RTD and ERP features does not have a similar distribution. The whole topographic maps of the correlations for both groups are presented in Fig S6.1 in (S6 Appendix).

The second CT feature corresponding to the mentioned criteria is final score in VS test. This feature has significant negative correlations with mean amplitudes for N200 during the

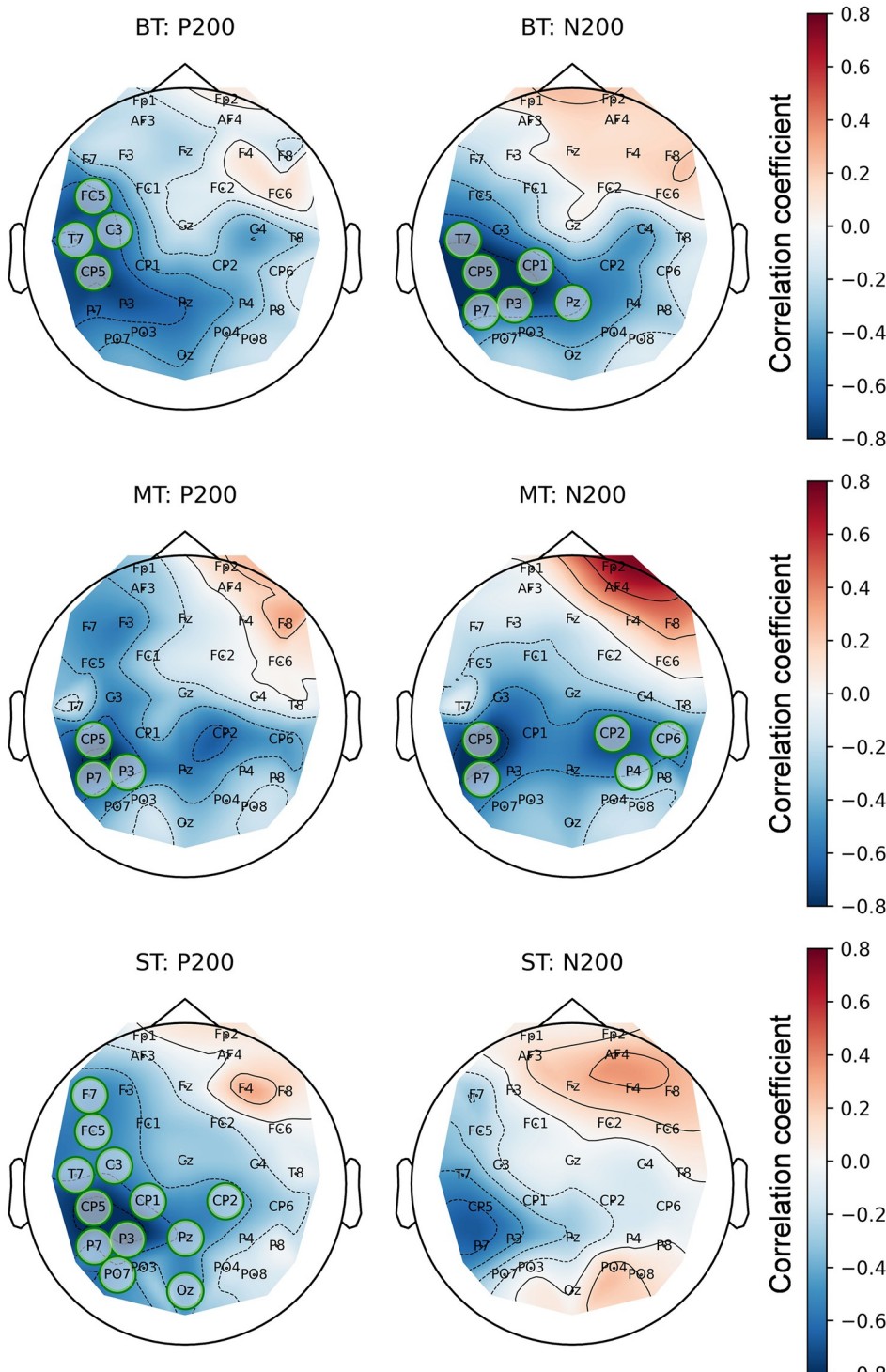

**Fig 11. Topographic maps of correlations between correct rate in RTD and mean amplitudes of ERP components for the NOVICE group.** Columns represent ERP components: P200 and N200. Rows represent EEG paradigms: Big Terrors (BT), Medium Terrors (MT), and Small Terrors (ST). The statistically significant channels are highlighted by green circles.

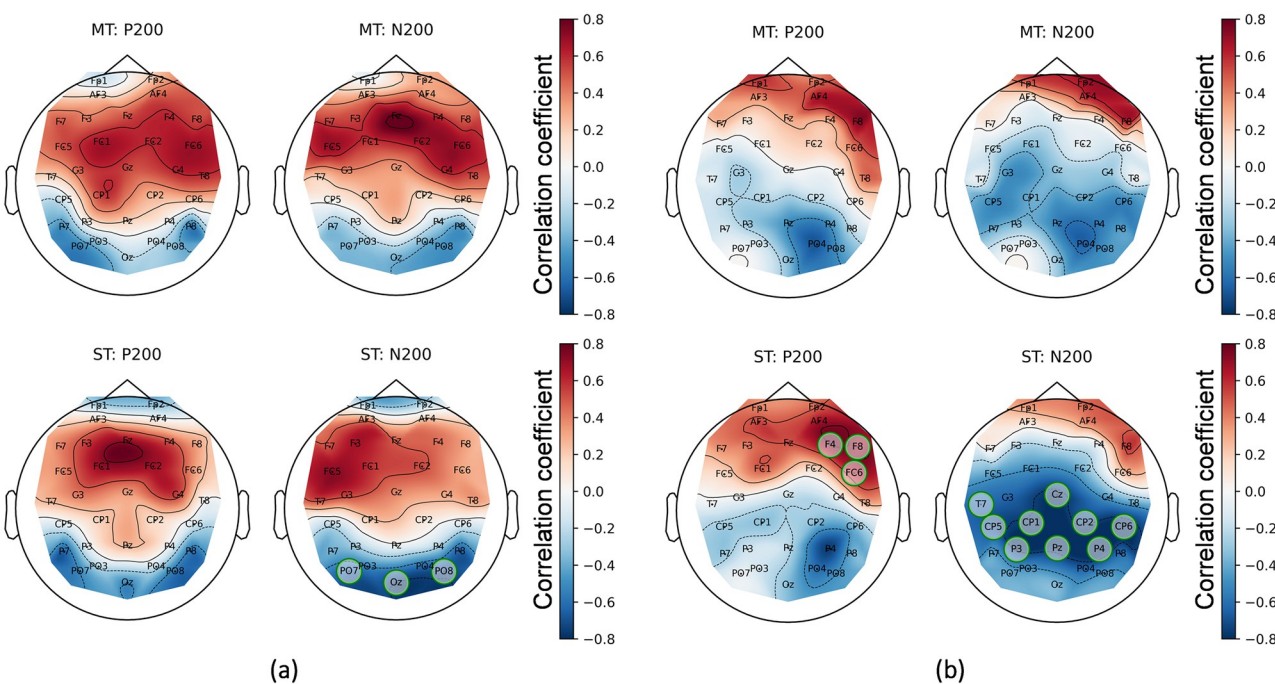

**Fig 12. Topographic maps of correlations between final score in VS and mean amplitudes of ERP components for PRO (a) and NOVICE (b) groups.** Columns represent ERP components: P200 and N200. Rows represent EEG paradigms: Medium Terrors (MT) and Small Terrors (ST). The statistically significant channels are highlighted by green circles.

Small Terror block of EEG experiments for professional and novice players (Fig 12). These correlations are primarily allocated in the occipital for the PRO group and parietal for the Novices group. It is worth noting that a similar correlation distribution but without channels with statistically significant correlations is also observed for the MT paradigm. This is consistent with that the Medium Terrors paradigm requires visual search skills to a much lesser extent than the Small Terrors paradigm. The whole topographic maps for both groups are presented in Fig S6.2 in (S6 Appendix).

As a preliminary summary, it is worth noting that only two features (Correct rate in RTD and final score in VS) from Cognitive Tests showed significant correlations with EEG features and only with mean amplitudes of ERP components. However, the statistical test results showed that the PRO and NOVICE groups are not significantly different from each other by these CT features.

## Correspondence of player's level with the CT results and ERP signals

No statistically significant correlation is observed between hours spent in CS:GO and CT features. The highest correlation is -0.31 with mean search time of "no L", which is low and not statistically significant (p = 0.38). In other words, the player in the PRO group who spent more time in the game did not have better results in RT and VS tests.

However, hours spent in CS:GO significantly correlate with mean amplitude of P200 and N200 during CS:GO-related blocks of EEG experiments. These correlations are presented in Fig 13. The whole topographic maps are presented in Fig S6.3 in (S6 Appendix). These correlations are mostly observed on the left side of the parietal cortex. The correlation allocations are similar for all mentioned ERP components and EEG paradigms, nonetheless, they are the

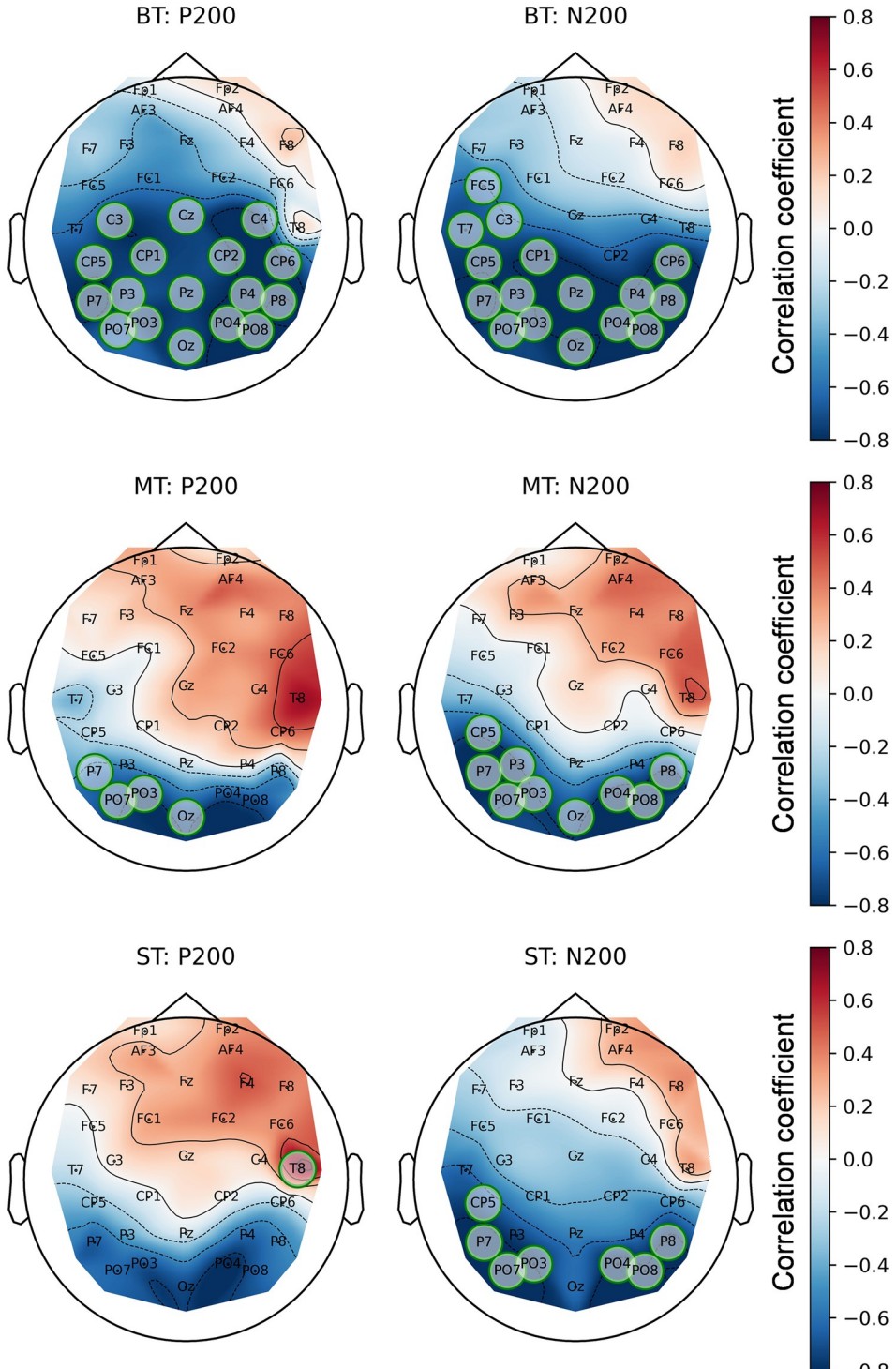

**Fig 13. Topographic maps of correlations between hours spent in CS:GO and mean amplitudes of ERP components for professional players.** Columns represent ERP components: P200 and N200. Rows represent EEG paradigms: Big Terrors (BT), Medium Terrors (MT), and Small Terrors (ST). The statistically significant channels are highlighted by green circles.

most visual for P200 during all game-related paradigms. It should be noted that this correlation distribution satisfies the established in the previous subsection criteria.

## Discussion

### Cognitive tests

The statistical analysis of the results of the cognitive tests on reaction time showed that the PRO group has better (shorter) reaction time for all considered variants of response actions (mouse click response action, keystroke response action, and decision-making) than the NOVICE group. Therefore, it can be concluded that professional players are superior to novice players in reaction time. This conclusion corresponds to previous studies [29–31]. However, it is worth mentioning that the difference in reaction time between professional and novice players in League of Legends was not detected in [37]. It can be related to the game genre. Apparently, as reported in [28], reaction skills are more important for First Player Shooter games like CS: GO than for MOBA games like League of Legends.

In addition, the professional players had significantly faster reaction time during a mouse click response action than keystrokes (see Fig 3(a)). Moreover, the correlation analysis within the PRO group showed a significant positive correlation between the reaction time during decision-making (choice reaction time) and reaction time during keystrokes (see Fig 9), which means a link between them. In other words, the reaction time of professional players with a mouse click as a response action differs from reaction times during other response actions. The NOVICE group, on the opposite, did not demonstrate a statistically significant difference between reaction time during mouse clicks and reaction time during keystrokes (see Fig 3(b)).

Moreover, the intragroup analysis for novices did not reveal any significant correlations between reaction time for all considered response actions (see Fig 10). This finding implies that playing in CS:GO may enhance the specific skill of fast mouse clicking, which is a key action in active shooter games, like CS:GO. This is consistent with the hypothesis that video game training can improve perceptual and motor skills [80]. However, this implication needs to be verified by further research.

As for reaction time during decision-making (choice reaction time), the results showed that the necessity of decision-making significantly increases the reaction time for both groups. However, the PRO group still showed a faster reaction. Taking into account the lack of statistically significant differences between the considered groups for the correct rate of RTD test, it is possible to conclude the following. Professional players can cope with a simple task (like distinguishing the color of the circles) during a time shortage faster than novice players with the same level of correctness. Thus, it is possible to conclude that professional players are superior to novice players in reaction time with decision-making. However, this superiority is related to better reaction but not better decision-making. It is worth noting that the authors got a similar result in [30]. They reported a statistically significant difference in switch cost reaction times, while the error rates for video-game players and no video-game players did not differ significantly.

In contrast to reaction time tests, the visual search test does not show statistically significant differences for any considered metrics between the PRO and NOVICE groups. In other words, the superiority of PRO players in general visual search skills was not observed. Thus, there is no evidence to confirm that professional players have better general visual search skills compared to novice players. This conclusion does not correspond to other research [41–43] that can be explained by the difference in the setup of visual search tests. The conducted Visual Search test is a test on the central search of the special letter ("L") among the distracting letters ("T"), while in the mentioned papers, more complex tests were considered (feature and

conjunction search, peripheral search, identification task while simultaneously performing a central search). In other words, the superiority of professional players could be revealed when performing more complex visual search tasks. It also should be noted that the PRO group demonstrated significantly less counting error of target stimuli during the Small Terrors block of the EEG Experiments. Hence, as expected, the professional players were better at solving a game-specific search task than the novice players.

## EEG experiments

As per ERP results, we observe significant differences between the considered groups in time latency (shorter of PRO group) and amplitude (higher for PRO group) of main ERP components (P200, N200, and P300) for game-related experiments—see Table 5. Similar differences in ERP peaks were previously reported with regard to traditional sports athletes (e.g., [8, 11, 13, 44]) and video game players [35] (see more S1 Appendix).

It is worth mentioning some cognitive functions that are reported [81] to be connected to ERP components. P200 may reflect the sensation-seeking behavior of an individual. N200 consists of 3 components: N2a / Mismatch negativity (MMN), N2b, and N2c. N2a/MMN represents the brain's automatic process involved in the encoding of the stimulus difference or change. N2b is slightly later in latency than N2a and appears when changes in the physical properties of the stimulus are task-relevant. N2c is the classification N2, elicited when the classification of disparate stimuli is needed. The latency of P300 is usually interpreted as the speed of stimulus classification resulting from the discrimination of one event from another. Shorter latencies indicate superior mental performance relative to longer latencies. P300 amplitude seems to reflect stimulus information such that greater attention produces larger P300 waves.

Interestingly enough, in the Blue/Red Ball paradigm, the PRO group did not show statistically significant differences w.r.t. NOVICE in amplitudes of ERP components (this can be seen in Fig 4), but we see the differences in latency in P200, N200, and P300 components (Table 5), which indicates faster visual reaction.

The significant difference only in P300 latency in the ST paradigm between the groups can be explained by higher visual target search time and its high deviation from person to person, while the visual search pattern of professional players (as can be seen from eye-tracker data—not presented in this paper) is quite stable both for a person and in the group.

With respect to spectral differences [82], reported stronger alpha event-related desynchronization over occipital and frontocentral sites occurred in experts compared to the novices when anticipating. As shown in Fig 5, professional players showed stronger pre-stimulus alpha power and continuing 200–300 ms after stimulus presentation, indicating higher levels of "quiescence, automaticity and fluidity" [83]. A number of studies conclude that the alpha rhythm can be considered a mechanism for increasing signal-to-noise ratios within the cerebral cortex by means of inhibition of unnecessary or conflicting processes to the current task [15, 84, 85].

We conclude that it is not the alpha suppression, which is higher for professionals after target stimuli presentation but rather their alpha power is higher during non-target presentation resulting in such significant power difference (see S4 Appendix). Higher alpha levels of professionals during rest (short time periods (250–500 ms) in between image presentations) are well explained by fluid intelligence theory and attention allocation. This hypothesis suggested that individuals with higher fluid intelligence tend to flexibly and adaptively allocate their limited resources according to the task type and task difficulty rather than simply exert more or fewer resources in any condition [86].

In addition to alpha power, the stronger beta power in occipital electrodes of professional players was also observed, which is reported to be connected with movement [87]. Our results

suggest that professional cybersportsman have stronger "shoot on sight" reflexes which exhibit themselves on spectrograms, especially in game-related experiments.

CP decomposition of Spatial-Temporal tensor supports higher P300-P200 (Fig 6(b)) amplitudes for the PRO group in game-related paradigms of the EEG experiments. It is worth noting that the maximum temporal component intensity of the first CPD component is about 300 ms and could correspond to basically ERP with a clear P300 peak close to 300 ms after the stimulus. This component does not show the statistical differences between the compared groups and could represent the basic P300 brain reaction to a stimulus (target or non-target). The second CPD component, in contrast, has a significantly higher intensity for professional players during CS:GO-related paradigms. Moreover, its maximum peak represented P300 is a little bit after 300 ms and becomes more time-stretched for the ST paradigm. The similar smoothing of the P300 peak can be seen in Fig 4. Thus, it can be assumed that the second component could represent the more conscious brain reaction which is affected by the links between the person's experience (like expertise in video games) and the presented stimulus (screenshot from the game).

CPD of the Spatial-Temporal-Spectral tensor is able to extract the components related to P300. More specifically, the spatial distributions of the first and second components correspond to P3b and P3a sub-components of P300 [88], which, according to [89], are related to task difficulty. Indeed we observe a significant difference between groups in all experiments for the P3b component that depicts cognitive workload [90] (Fig 7(a)). P3a is significantly different only in MT and ST experiments as shown in Fig 7(b), 3rd and 4th columns. Although our experiments consisted of a target (red ball or terror) and standard (blue ball or empty map) stimuli without rare nontarget required for P3a elicitation, one can argue that, in the MT and ST paradigms, the participant needs to spend some time on visual search task before being able to certainly classify the image as consisting terror or empty.

CPD of Spatial-Temporal-Spectral tensor in higher frequencies shows alpha suppression in the posterior region after stimulus presentation, but it is not statistically different between groups (Fig 8 column 2). The result is in accordance with spectrogram analysis (see in Fig 5).

## Intergroup and intragroup relations between cognitive tests and ERP signals

The following conclusions can be made from the joint analysis of the EEG experiments and the Cognitive Tests. As for reaction time skills, the better (faster) reaction time of the professional players (compared with the novices) for different types of tests (with mouse click response action, keystroke response action, and decision-making) can evidence that the superiority of professional players in reaction time is related not only to faster response actions but also with the faster cognitive processing. The fact that the ERP peaks (P300, N200, and P200) are observed for professional players earlier (statistically significant) than for novice players for all blocks of EEG experiments, including the non-specific Blue/Red Ball (B/RB) paradigm, confirms this statement. In other words, CS:GO game can have a positive influence on reaction skills, which is related to changes (faster ERP responses) in brain activity like a professional sport does, according to numerous studies [7–9, 11–14].

However, the intragroup analysis within the professional players did not show significant correlations between the reaction time during different cognitive tests and ERP features. It is worth noting that in contrast to other research [8, 44, 46, 71], we considered average values of the cognitive tests results and EEG features but not the ERP signals during cognitive tests. Hence, the participant, who demonstrated a shorter average reaction time during cognitive tests, can show a lower average amplitude and longer average latency of ERP peaks than

another participant with a worse average reaction time. In addition, the analysis within the professional players' group between hours spent in the game and the reaction times also revealed a non-significant correlation. Thus, the superiority of professional players in reaction time to novices is mostly related to intergroup factors, i.e., the difference between professional and novice players. In other words, the fast reaction can be the necessary cognitive ability for eSport players in CS:GO, which does not have a strong linear link to their immersion and experience in the game.

Another important finding is that a different effect of the connection between the EEG paradigm and CS:GO for the considered groups is also observed in relations between Cognitive tests and ERP signals. It is worth noting that the participant should make a decision (chose) if the presented picture is a target stimulus (e.g., there is a terrorist) or not in the EEG experiments. It is also required in the cognitive test on reaction time during decision-making (choice reaction time). The analysis showed that the Novice group has a similar correlation distribution between mean amplitude of ERP peaks and correct rate during the cognitive test on choice reaction time (reaction time during decision-making) for all EEG paradigms, regardless of whether the paradigm is related to CS:GO or not. In contrast, for the PRO group, the correlations have a different distribution picture for Blue/Red Ball paradigms than for the CS:GO-related paradigms (see Fig S6.1 in S6 Appendix). Moreover, there is a relatively high positive correlation of combined ERP features (a mean value of features for Fz, Cz, and Pz channels) between paradigms related to CS:GO for the professional players in contrast to the novices (see Figs 9 and 10).

We want to point out that the correlations between the final score in the visual search test and the N200 ERP amplitudes (see Fig 12) show that ERP can reflect visual search skills in both groups. We found significant negative correlations in the occipital (for the PRO group) and parietal (for the NOVICE group) regions for the N200 ERP component for the ST paradigm, which involves visual search. This is in line with other studies (like [91–93]) that investigated the brain's visual search mechanism using ERP from the same head areas. The main difference from the previous studies is that we observed a significant correlation even when the visual search tests were done separately from the EEG experiments. The absence of evidence for the superiority of professional players in visual search skills, on the other hand, matches the fact that we saw the same correlation pattern for both professional players and novices.

We can also conclude that visual search skill is not the only factor that affects the performance in finding the terrors on the CS:GO pictures during the EEG paradigm with Small Terrors. It may also depend on the experience (search strategy) of professional players (compared to novices) in the game. The higher amplitude of the P300 ERP component for the professional players (as an indicator of immersion in the game) than for the novices supports this idea.

It should be noted that the detected significant correlations between ERP and Cognitive Tests features are allocated in the occipital and parietal areas of the head. This situation is explained by the fact that these regions correspond (occipital) or are close (parietal) to the visual cortex of the brain, which is activated during a response to visual stimuli.

## Correspondence of player's level to the cognitive tests and ERPs

In this section, we verify the relations between hours spent in CS:GO with the CT and ERP features. The absence of a statistically significant correlation between the cognitive tests and hours spent in the game can indicate that first-player shooters (like CS:GO) do not have a strong linear link with reaction time or visual search skills. In [28], the authors tested more than 35 people per group and also did not find a significant correlation between hours of

eSports per week and reaction times. However, in [94], the statistically significant correlation between "Duration of regularly playing video games in years" and reaction time based on data from 432 participants was reported. We should note that the absolute values of these correlations were less than 0.16, which is too low to be reliably detected with 10 participants per group. The non-significant correlation between the visual search features and the hours in the game, in turn, was expected because there was no statistical difference in visual search features between professional and novice players. Similarly [37], showed no significant correlation between visual search results and player's rank.

The significant negative correlations between hours spent in the game and ERP features (mainly P200 and N200 mean amplitudes) suggest that ERP signals are related to the skill level of CS:GO players. This is consistent with previous studies on professional athletes that reported similar findings [6, 8, 44–46]. The localization of these correlations in mostly the left occipital, parietal, and temporal cortexes can be attributed to two factors. First, all the participants in the PRO group were right-handed, which implies a more active left hemisphere. Second, the participants were exposed to visual stimuli during the EEG experiments, which naturally activated the regions near (parietal) or corresponding to (occipital) the visual cortex.

## Associating eSport with a real sport

We want to emphasize the similarities between real sports and eSports. It was observed that professional athletes have superior cognitive skills related to their sport than the control group (e.g., footballers [23] and orienteering athletes [24] in problem-solving skills; tennis players [25] in inhibitory control and cognitive flexibility). An analogous superiority in reaction time skills was detected for professional CS:GO players in the current research as well as in previous research, e.g., [28, 29, 37]. Besides cognitive skills, differences between high-level athletes were revealed in EEG brain activity for both temporal and frequency domains, as was shown in several reviews [4, 6]. For instance, the differences in amplitude and latency in P300 were revealed for athletes and the control group in [7, 11]. Similar distinctions in P300 were detected in the current research between the professional players and the novices (control group). It is worth noting that [37] also showed the difference in brain reactions of players of various levels. The authors concluded that professional, trained, and novices players can be discriminated by EEG signals during a game. Thus, based on the written above, the current research can serve as evidence that eSport has an influence not only on cognitive skills but on brain responses too, as real sports do for athletes. In other words, this study supports the opinion that eSport can be considered one of the sports.

## Conclusion

In this section, we make conclusions about the hypotheses stated in the introduction section and the practical influence of the current research.

1. The hypothesis about the differences between professional and novice players in CS:GO manifesting in spectral and temporal characteristics of the EEG signal was confirmed. In the temporal domain, a significant differences (shorter latency and higher amplitude of the ERPs) were observed in visual representation, statistical tests, and tensor-based analysis. In the frequency domain, the professional players demonstrated significantly stronger stimulus-locked alpha-band power. We assume that it is mainly related to the higher alpha power of the professional player during rest periods but not their greater alpha suppression.

2. The hypothesis about the expected differences between the considered groups was fully affirmed. The professional players showed higher ERP amplitude of ERP peaks for all CS: GO-related EEG paradigms while not for the non-specific paradigm. It can mean a higher engagement of professional players in the CS:GO game. Besides, the professional players demonstrated faster reactions expressed in significantly shorter latencies of the ERPs and shorter reaction times during the cognitive tests.

3. The hypothesis about the superiority of professional players to novices in the considered cognitive skills was partially confirmed. The professional players had a significantly shorter simple reaction time. However, the difference in visual search skills and choice correctness was not statistically significant.

4. The hypothesis about the expression of studied cognitive skills in the participant's brain activity was partially affirmed. The cognitive tests on reaction time showed the intergroup (but not intragroup) relation with ERP components. The visual search test, in contrast, had non-significant intergroup differences when the significant intragroup negative correlation with the ERP amplitude was observed for both considered groups.

5. The fifth hypothesis stated that Cognitive tests and EEG experiments could be used to detect the level of the CS:GO player (based on hours spent in the game) and was partially confirmed. The correlation between the hours spent in the game and ERP features supports that the level of the CS:GO player can be detected based on the amplitude of the ERP components.

As for the practical influence of the research, cognitive tests on simple reaction time and choice reaction time can be used to separate professional and novice players by their cognitive skills. This separation can facilitate the selection of the most suitable training plan. The EEG evaluation, in turn, can be used to rank the level of the players more accurately in order to select the most potential players.

Samples of the data used in our study, as well as an example of analysis in the paper, are available at https://disk.yandex.ru/d/jgW45J3tJKUAWw.

## Supporting information

**S1 Appendix. Summary table of EEG studies.**
(PDF)

**S2 Appendix. Summary table of cognitive skills studies.**
(PDF)

**S3 Appendix. Cognitive tests description.**
(PDF)

**S4 Appendix. Topographical spectrums.**
(PDF)

**S5 Appendix. ERP components peak amplitude and latency distribution.**
(PDF)

**S6 Appendix. Correlation analysis.**
(PDF)

**S1 Data.**
(ZIP)

## Acknowledgments

We thank all members of the Kronbars Esport team of ITMO University (St. Petersburg) who participated in our experiments as a member of the PRO group and all participants who took part in our experiments as a member of the NOVICE group. Additional thanks to Alexandra Razorenova for her comments on EEG preprocessing and visualization. We also would like to show our special gratitude to the students from ITMO University who facilitated our experiments. Primarily, we would like to present our special thanks to Mikhail Gerasimov for organizing communication with the Kronbars team and Nikita Agrazhenkov, the Kronbars team manager, who perfectly organized our experiments at ITMO University.

## Author Contributions

**Conceptualization:** Andrei Znobishchev, Anh-Huy Phan, Andrzej Cichocki.

**Data curation:** Sergei Gostilovich, Airat Kotliar Shapirov, Andrei Znobishchev.

**Formal analysis:** Sergei Gostilovich, Airat Kotliar Shapirov, Andrei Znobishchev.

**Investigation:** Sergei Gostilovich, Airat Kotliar Shapirov, Andrei Znobishchev.

**Methodology:** Anh-Huy Phan.

**Project administration:** Anh-Huy Phan.

**Supervision:** Anh-Huy Phan, Andrzej Cichocki.

**Validation:** Sergei Gostilovich.

**Visualization:** Sergei Gostilovich, Airat Kotliar Shapirov.

**Writing – original draft:** Sergei Gostilovich, Airat Kotliar Shapirov, Andrei Znobishchev, Anh-Huy Phan.

**Writing – review & editing:** Sergei Gostilovich, Airat Kotliar Shapirov, Anh-Huy Phan.

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
