## [Decision Letter · Decision Letter 0]

19 Dec 2022

PONE-D-22-26460Biomarkers of Professional Cybersportsmen: Event Related Potentials and Cognitive Tests StudyPLOS ONE

Dear Dr. Gostilovich,

Thank you for submitting your manuscript to PLOS ONE. After careful consideration, we feel that it has merit but does not fully meet PLOS ONE’s publication criteria as it currently stands. Therefore, we invite you to submit a revised version of the manuscript that addresses the points raised during the review process.

The manuscript is overall well-written except that some of the parts can be further improved. Please revise the manuscript substantially according to the reviewers' comments, such as including details in the description of methods, improving the quality of figures, etc.

We look forward to receiving your revised manuscript.

Kind regards,

Chun-Shu Wei

Academic Editor

PLOS ONE

Journal Requirements

Additional Editor Comments:

The manuscript is overall well-written except that some of the parts can be further improved. Please revise the manuscript substantially according to the reviewers' comments, such as including details in the description of methods, improving the quality of figures, etc.

Reviewers' comments:

Reviewer's Responses to Questions

**Comments to the Author**

1. Is the manuscript technically sound, and do the data support the conclusions?

Reviewer #1: Yes

Reviewer #2: Yes

2. Has the statistical analysis been performed appropriately and rigorously? 

Reviewer #1: Yes

Reviewer #2: Yes

3. Have the authors made all data underlying the findings in their manuscript fully available?

Reviewer #1: Yes

Reviewer #2: No

4. Is the manuscript presented in an intelligible fashion and written in standard English?

Reviewer #1: Yes

Reviewer #2: Yes

5. Review Comments to the Author

Reviewer #1: Authors investigated the differences in EEG biomarkers and behavioral performance between professional FPS gamers vs non-gamers. This topic is interesting and quite important in esports area.

Also, this paper is well-organized and provided all technique details for readers. I was wondering how ERPs could be collected if they were using commercial FPS games (e.g., counter strike here) instead of customized games. Ultimately, I understand that they made a random sequence of game screenshots of interest! This is brilliant. Also, one thing is impressive is that they used photodiode to syncronise the EEG signals and visua stimuli, which is a standard and rigorous way for ERP studies but always overlooked by other researchers.

For EEG data analysis, they employed EMD to calcualte spectrogram instead of conventional FFT; also they investigated the specific main waves and frequencies with tensor decomposition. All these data analysis methods are the current state-of-the-art. But my suggestion is that since professional game players are well-trained people in multitasking FPS cognitive enviornment, so brain network-based functional connectivity analysis is perhaps better in capturing their unique brain actitivity patterns in comparison with non-gamers. So, this is something authors can take into consideration in their revision.

In addition, I understand that authors just used a set of game screenshots in their study rather than real CS game, but i am still curious that for those non-gamers, did they experience any cybersickness or motion sickness when they watched those screenshots? If so, how cybersickness or motion sickness affected their EEG biomarkers or cognitive test performance?

Reviewer #2: Comments to the authors:

The authors aims to find the biomarkers to distinguish professional and novice video game players players. The manuscript is well written and organized. However, the resolution of all of the figures need to be improved so that the labels are readable. Moreover, some of the figures labels are missing. I have some remarks for the authors.

1. In the Section Motivation, the authors state the contributions of the manuscript are understanding the influence of eSport on a player's cognitive state and associating eSport with a real sport. Though the authors have provided sufficient results from previous studies, I suggest the authors to put all the previous findings into a table. Thus, the impact of the manuscript can be emphasized (According to the table, none of the reseach has use ERP to analyze FPS gamers, etc.) and the readers can have an expection about how the metrics change (reaction time, multi-task tracking, etc.) in the following Sections.

2. In line 267, the authors introduce the CS:GO-related EEG experiment. During each condition, is the location of ther target fixed? If the target location is fixed, the size of the target might have little effect and visual search process might not be necessary, which change the proposed metrics from choice reaction time to visual attention only.

3. In line 325, the authors use ICA to remove blink and ECG artifacts. However, only blink removal has been explained in the following paragraph but not ECG. Further explanation needed on how the authors remove ECG artifacts.

4. In line 370, it seems that the sentence is not completed: on the other hand, can (?) [49] and has been...

5. In Section ERP (line 502) and Fig. 4, it seems that the authors only plot the deviant trials. I suggest the authors to plot out standard trials as well and see if the results support the statement in line 524: the appearance timing of almost all components of professionals is approximately 1-0-40ms earlier than the novice, and the statement in line 702: Pro group has faster visual reaction.

6. In Fig. 5, there is an abnormally high power from 0Hz to 10Hz across the whole trials in the Pro group. One possible reason is that the Pro players are continuously moving their eye gaze to search for the targets and generating these low-frequency artifacts. Once again, this question goes back to my question 2.

7. In Fig. 6, though the authors have described the axis-labels in the caption, it will be more reader-friendly if the axis-labels are on the figure. Also, the resolution of the figure is too poor to read the legend.

8. Following my question 6 and 7, it seems that CPD component 1 might explain why there is a significant increase in low-frequency power in the Pro group. I suggest the authors to separate the trials by groups and recalculate the power once again.

9. In Fig. 6, component 2 has response right before stimuli onset. Does it mean subjects expect there is an incoming stimuli and begin to prepare for the potential deviant trial? Are the component 2's activities same between two groups? Also, does CPD include only deviant trials or all the trials? The statement in line 732 is potentionlly weaken by these questions.

10. Out of my personal curiousity, why there is no EEG recordings during cognitive tests? I expect there is a difference in the baseline before stimuli onset between Pro and novice players.

Some minor suggestions:

1. In line 41, EMG was used without definition. I suggest the authors to add it in line 39: [2] investigated electromyography (EMG), the electrical activity of muscle tissue, of lower....

2. In line 57, EGG was used without definition. I assume it is a typo for EEG.

3. In line 205 and line 220, the authors seem to make the same claim while refereing to different references (6 in 205 and 4 in 220).

4. In line 433, PLS and CCA were used without definitions.

5. In line 720, Gf was used without definition.

6. PLOS authors have the option to publish the peer review history of their article (what does this mean?). If published, this will include your full peer review and any attached files.

Reviewer #1: **Yes: **GANG LI

Reviewer #2: **Yes: **Chi-Yuan Chang

---

## [Author Response · Author response to Decision Letter 0]

14 Mar 2023

Please, find the response in the attached file "Response to reviewers.pdf".

---

## [Decision Letter · Decision Letter 1]

27 Apr 2023

PONE-D-22-26460R1Biomarkers of Professional Cybersportsmen: Event Related Potentials and Cognitive Tests StudyPLOS ONE

Dear Dr. Gostilovich,

Thank you for submitting your manuscript to PLOS ONE. After careful consideration, we feel that it has merit but does not fully meet PLOS ONE’s publication criteria as it currently stands. Therefore, we invite you to submit a revised version of the manuscript that addresses the points raised during the review process.

We look forward to receiving your revised manuscript.

Kind regards,

Chun-Shu Wei

Academic Editor

PLOS ONE

Additional Editor Comments :

The manuscript is overall well-written except that some of the parts can be further improved. Please revise the manuscript substantially according to the reviewers' comments.

Reviewers' comments:

Reviewer's Responses to Questions

**Comments to the Author**

1. If the authors have adequately addressed your comments raised in a previous round of review and you feel that this manuscript is now acceptable for publication, you may indicate that here to bypass the “Comments to the Author” section, enter your conflict of interest statement in the “Confidential to Editor” section, and submit your "Accept" recommendation.

Reviewer #2: All comments have been addressed

Reviewer #3: (No Response)

2. Is the manuscript technically sound, and do the data support the conclusions?

Reviewer #2: Yes

Reviewer #3: Partly

3. Has the statistical analysis been performed appropriately and rigorously? 

Reviewer #2: Yes

Reviewer #3: Yes

4. Have the authors made all data underlying the findings in their manuscript fully available?

Reviewer #2: No

Reviewer #3: No

5. Is the manuscript presented in an intelligible fashion and written in standard English?

Reviewer #2: Yes

Reviewer #3: No

6. Review Comments to the Author

Reviewer #2: I appreciate the authors' efforts. Most of my questions have been properly addressed. One last thing, I strongly recommend the author, for reference [4] ICLabel, to cite the paper instead of website. The paper title is "ICLabel: An automated electroencephalographic independent component classifier, dataset, and website".

Reviewer #3: In this study, the authors investigated the differences in EEG and cognitive test performance between eSport PRO and NOVICE players. The results showed that PRO players had significantly faster reaction times and stronger and earlier event-related potentials (ERPs), as well as stronger pre-stimulus alpha power, compared to NOVICE players. Visual search performance, however, did not differ significantly between the two groups. Using tensor decomposition analysis, the authors found that the second component of the p300-like component was significantly stronger in PRO than NOVICE. Furthermore, correlation analysis revealed different patterns between PRO and NOVICE players, with PRO players showing higher correlations between ERP features in CS:GO-related paradigms, while NOVICE players did not.

Overall, this study provides valuable insights into the neural and cognitive correlates of eSports professionals. The authors thoroughly explored potential factors that may differ between PRO and NOVICE players. However, there are a few major conceptual and analysis issues, as well as additional clarity, that need to be addressed before this study can reach the required bar for influencing thinking in the field. Nonetheless, the study's findings are interesting and contribute to the ongoing research on eSports performance.

General comments

1. The authors presented a comprehensive analysis of the EEG and cognitive test performance differences between eSport PRO and NOVICE players. To improve the clarity of the paper, it would be beneficial for the authors to prioritize the most significant findings and clearly link them to the hypotheses listed in the Introduction.

2. Furthermore, some of the conclusions drawn by the authors appear to be unsupported or even contradicted by the empirical findings. It is important for the authors to carefully examine their results and revise their conclusions accordingly. Additionally, they should aim to provide more clarity in describing and interpreting their results.

3. One area that could be further developed is the discussion of the cognitive functions associated with the brain components investigated in this study, such as the ERP P200, P300, N200, and alpha-band oscillations. It would also be helpful for the authors to discuss how these components are related to each other and to the cognitive demands of eSport gameplay.

Detailed comments

Abstract: do the authors intend to say pre-stimulus alpha instead of post-stimulus alpha? In Figure 5 and corresponding main text from Line 561, as well as Discussion described the effects otherwise.

Line 53: It is better to spell out what are some early/late components and which ones do authors focus on and why.

Line 79 & Line 571: define the frequency range of theta, alpha and beta

Line 118: “faster” in what aspect?

Line 180: it’s unclear about what “representation” mean in this context. Do the authors mean the underlying neural processes of a certain cognitive function?

Line 227: please provide more context to motivate EMD analysis

Line 251: for a more balanced literature review, please also provide counter-evidence of the benefits of eSports if there is any

Line 234: were the CS:GO screenshots directly extracted from the game? Plus that PROs brought their own equipment. One might argue that the superior performances (e.g. faster reaction times) of the PRO may be due to familiarity of the scene and better equipment.

Line 397: please be more clear about why ICA/PCA cannot take into consideration of multi-dimensional data.

Line 426: it would be beneficial for the authors to have a summary table here including all result parameters from different CT and EEG tests.

Line 452: what tasks did participants do in the reaction time tests?

Line 457: how was the threshold determined?

Line 469: incomplete sentence

Line 504: “In other words, it obviously implies that professional players have a faster reaction time than novice players.” is redundant

Fig 2: in my view of the figure, there were no abcde labeled. Please check for all other figures too. In addition, please add the units for all the subplots for this figure and all other figures.

Table 3 and corresponding statistics in the main text: it would be important to add the effect size since p-value is very easily biased by the number of the datapoints. Is there a reason why only showed ST but not BT and MT conditions?

Fig 3: (c) (d) labels were inaccurate – should be RTM & RTD

Line 527: the authors mentioned that VS test was not related to CS:GO game directly. Was the RT tests related to CS:GO directly?

Fig 4: How were the channels selected? Make sure the y-axis scale is the same.

Line 541: the word “latency” and “onset time” were used interchangeably. I suggest use consistent terms throughout the paper.

Table 4: why only showing Pz instead of the average as in the Figure 4?

Fig 5: should be “spectrograms” instead of “ERP spectrogram”

Line 578: how “high” is the correlation? Please be clear about the cut-off points for selecting components.

Fig 6: incomplete sentence “All paradigms were grouped the number of the component.”

Line 610: what is the third component?

Line 612: suggest consistent terms such as “occipital“ instead of “back part of the head”

Line 625: 3 to 30 Hz?

Line 642: “CS:GO-related paradigms are paradigms are emphasized from the non-specific paradigm“ meaning “correlations are higher in CS:GO related paradigms”? Please rephrase to be clearer.

Line 656: “this correlation does not pass the 5% threshold of p-value“ can simply be rewritten as “not significant”

Line 665: it would be awesome if the authors could highlight the ones based on those two criteria

Fig 11 -13: please interpret why the effects are stronger in these certain channel locations in Discussion.

Line 708: one general comment on the writing style of the paper is that it would be more informative to report the direction of the effect rather than simply stating that there is a difference between PRO and NOVICE players. The authors could provide more clarity by specifying whether the effect is stronger or weaker in one group compared to the other.Line 721: please consider rephrasing this sentence to more clearly contrast which correlations are higher in PRO vs. NOVICE

Line 726: meaning fewer “significant” correlation?

Line 730: the finding of faster mouse clicking could be due to the equipment that PROs brought with them. It would be helpful to know the baseline clicking rates of PRO and NOVICE.

Line 761: please expand and explain the findings of those ERP components and they can relate to each other.

Line 764: “PRO group does not show statistically significant differences” from NOVICE?

Line 768: spell out what are lower significant differences. Are these shown in Table 4?

Line 772: both pre- and post-stimulus alpha-band power were significantly stronger in PRO than NOVICE in Figure 5, why only mention pre-stimulus alpha power? This is also inconsistent with the abstract saying the “post-stimulus alpha”

Line 779: aren’t the difference between PRO vs. NOVICE instead of target vs. non-target? Please explain.

Line 823: how can the statement be claimed based on Line 825’s non-significant results?

Line 824: spell out the “change” e.g. faster responses

Line 833: could expand on this point, so where does this superiority of PRO in reaction time comes from?

Line 848: what was the new findings in addition to previous VS tests?

Line 859: if this statement about experiences is true, the authors should observe a correlation between hours of play and ERP amplitude.

Line 872: consider rephrasing “The absence of a significant correlation between the hours in the game and visual search features, in turn, is expected as the difference between professional and novice players was not observed.“ for clarity.

Line 881: it shouldn’t be titled as the Conclusion since it did not conclude anything about this study.

Line 900: please explain the benefits of separating PRO and NOVICE and using EEG to rank the players.

7. PLOS authors have the option to publish the peer review history of their article (what does this mean?). If published, this will include your full peer review and any attached files.

Reviewer #2: **Yes: **Chi-Yuan Chang

Reviewer #3: **Yes: **Tzu-Han Zoe Cheng

---

## [Author Response · Author response to Decision Letter 1]

1 Jun 2023

The full response to the Reviewers is uploaded in the corresponding file ("Response to reviewers.pdf")

---

## [Decision Letter · Decision Letter 2]

25 Jun 2023

PONE-D-22-26460R2Biomarkers of Professional Cybersportsmen: Event Related Potentials and Cognitive Tests StudyPLOS ONE

Dear Dr. Gostilovich,

Thank you for submitting your manuscript to PLOS ONE. After careful consideration, we feel that it has merit but does not fully meet PLOS ONE’s publication criteria as it currently stands. Therefore, we invite you to submit a revised version of the manuscript that addresses the points raised during the review process.

The authors are encouraged to address the identified issues and revise the manuscript accordingly.

We look forward to receiving your revised manuscript.

Kind regards,

Chun-Shu Wei

Academic Editor

PLOS ONE

Additional Editor Comments:

The authors are encouraged to address the identified issues and revise the manuscript accordingly.

Reviewers' comments:

Reviewer's Responses to Questions

**Comments to the Author**

1. If the authors have adequately addressed your comments raised in a previous round of review and you feel that this manuscript is now acceptable for publication, you may indicate that here to bypass the “Comments to the Author” section, enter your conflict of interest statement in the “Confidential to Editor” section, and submit your "Accept" recommendation.

Reviewer #3: All comments have been addressed

2. Is the manuscript technically sound, and do the data support the conclusions?

Reviewer #3: Partly

3. Has the statistical analysis been performed appropriately and rigorously? 

Reviewer #3: Yes

4. Have the authors made all data underlying the findings in their manuscript fully available?

Reviewer #3: No

5. Is the manuscript presented in an intelligible fashion and written in standard English?

Reviewer #3: No

6. Review Comments to the Author

Reviewer #3: The authors made a thorough effort to address my previous comments. They included explicit descriptions, tables, supplementary files, and relevant literature, resulting in a significantly enhanced manuscript that exhibits improved clarity in its description, methodology, and presentation of results. Nonetheless, there is room for further improvement in terms of conciseness and overall refinement of the paper. Please see my comments below.

1. Please carefully read through the paper to ensure the sentences are correct and easy to follow. For example, Line 633 “alpha power before and 200-300ms after the stimulus“ is grammatically incorrect.

2. How about changing the (mostly) pre- and post-stimulus alpha-band power to stimulus-locked alpha-band power?

3. In the Abstract, please rephrase “The comparison of cognitive tests showed statistically significant superiority of professional players in reaction time (faster) and choice reaction time - faster reaction, but similar correctness, while a significant difference in visual search skills was not detected.” since the superiority technically could not be statistically significant. I suggest report the results as the metrics you used first, then interpret the results.

4. Line 93: Thanks for adding the negative evidence to make the paper more complete. However, connection between GD and other literatures reviewed in the Introduction, especially in the previous and the next paragraph, should be further discussed.

5. Line 197: Need to rephrase the first two sentences. The authors reply to my question about representation; however, it is still not clear in the text. To the best of knowledge, representation is the neural activities that reflect certain cognitive or behavioral processes.

6. Line 244: Suggest adding temporal domain “of the neural signals” for the first hypothesis.

7. Line 516: I don’t think the parenthesis is needed.

8. Figure 3: there is an extra blank in 100 000 permutations.

9. Line 675 & Figure 8: It should be high-er frequency band because 3-30 Hz is not really considered “high-frequency”.

10. Line 805: What does it mean by “diverse”?

11. Line 813: I suggest change the “facts” to findings and “can mean” to “suggest”. Please make sure to discuss the findings within the scope of the current.

12. Line 850: “Similar” to “Similarly”?

13. Line 895: “Our suggestion is” to “Our results suggest”

14. Line 958: This sentence is circular, please rephrase.

15. Line 966: Please clarify “has another picture”

16. Line 1055: Please be careful about the distinction between “spectrum” and “spectrogram”. I believe the authors intend to say “spectrogram” as in the Figure 5, please make sure to use correct term throughout the paper. E.g. either take out Figure 5 or the spectrum of the ERPs

17. Line 1046: The Conclusion (and other parts of the paper) needs to be more precise. Please clearly state the hypothesis in one sentence, and the corresponding results in the next sentence for each point. For example, Hypothesis 4 first few sentences could be shortened to “We only analyze the relations between the cognitive and ERPs since they were recorded separately.” The details should not be included in the Conclusion.

7. PLOS authors have the option to publish the peer review history of their article (what does this mean?). If published, this will include your full peer review and any attached files.

Reviewer #3: **Yes: **Tzu-Han Cheng

---

## [Author Response · Author response to Decision Letter 2]

28 Jun 2023

We have responded to all the reviewers' comments in the attached file: "Response to reviewers.pdf"

---

## [Editor Report · Decision Letter 3]

17 Jul 2023

Biomarkers of Professional Cybersportsmen: Event Related Potentials and Cognitive Tests Study

PONE-D-22-26460R3

Dear Dr. Gostilovich,

We’re pleased to inform you that your manuscript has been judged scientifically suitable for publication and will be formally accepted for publication once it meets all outstanding technical requirements.

Kind regards,

Chun-Shu Wei

Academic Editor

PLOS ONE

Additional Editor Comments (optional):

This has been recommended for acceptance following the review process. Congratulations on the significant improvements you have made to the manuscript.
---

## [Editor Report · Acceptance letter]

21 Jul 2023

PONE-D-22-26460R3 

Biomarkers of Professional Cybersportsmen: Event Related Potentials and Cognitive Tests Study 

Dear Dr. Gostilovich:

I'm pleased to inform you that your manuscript has been deemed suitable for publication in PLOS ONE. Congratulations! Your manuscript is now with our production department. 

Kind regards, 

on behalf of

Dr. Chun-Shu Wei 

Academic Editor

PLOS ONE